# Let LRMs Break Free from Overthinking via Self-Braking Tuning

**Haoran Zhao**[1,2,*]    **Yuchen Yan**[1,*]    **Yongliang Shen**[1,†]    **Haolei Xu**[1]    **Wenqi Zhang**[1]

**Kaitao Song**[3]    **Jian Shao**[1]    **Weiming Lu**[1]    **Jun Xiao**[1]    **Yueting Zhuang**[1]

[1] Zhejiang University, [2] Tianjin University, [3] Microsoft Research Asia
`ran159753@tju.edu.cn`, {`yanyuchen, syl`}`@zju.edu.cn`

GitHub: `https://github.com/ZJU-REAL/Self-Braking-Tuning`
Project: `https://zju-real.github.io/SBT/`

## Abstract

Large reasoning models (LRMs), such as OpenAI o1 and DeepSeek-R1, have significantly enhanced their reasoning capabilities by generating longer chains of thought, demonstrating outstanding performance across a variety of tasks. However, this performance gain comes at the cost of a substantial increase in redundant reasoning during the generation process, leading to high computational overhead and exacerbating the issue of overthinking. Although numerous existing approaches aim to address the problem of overthinking, they often rely on external interventions. In this paper, we propose a novel framework, **Self-Braking Tuning** (SBT), which tackles overthinking from the perspective of allowing the model to regulate its own reasoning process, thus eliminating the reliance on external control mechanisms. We construct a set of overthinking identification metrics based on standard answers and design a systematic method to detect redundant reasoning. This method accurately identifies unnecessary steps within the reasoning trajectory and generates training signals for learning self-regulation behaviors. Building on this foundation, we develop a complete strategy for constructing data with adaptive reasoning lengths and introduce an innovative braking prompt mechanism that enables the model to naturally learn when to terminate reasoning at an appropriate point. Experiments across mathematical benchmarks (AIME, AMC, MATH500, GSM8K) demonstrate that our method reduces token consumption by up to 60% while maintaining comparable accuracy to unconstrained models.

## 1 Introduction

Large reasoning models (LRMs) such as OpenAI's o1 [3], Deepseek-R1 [4], QwQ [5], Gemini 2.0 Flash Thinking [6] and Kimi-1.5 [7], excel at mathematical and logical tasks by generating detailed multi-step reasoning, boosting accuracy on complex benchmarks [8]. However, this often results in excessively long inference trajectories, frequently consuming thousands of tokens per problem [9, 10],

---

[*] The first two authors have equal contributions. This work was done when the first author was an intern at Zhejiang University.
[†] Corresponding author.

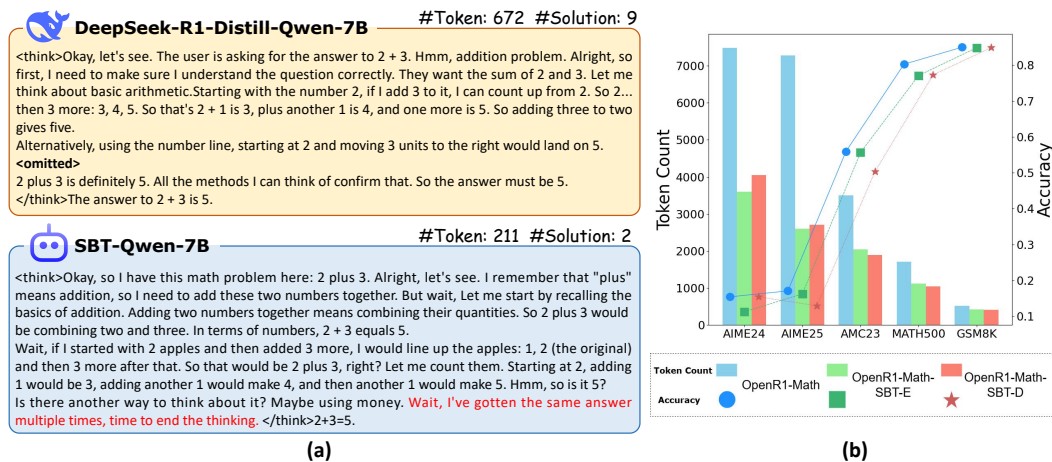

Figure 1: Demonstration of Self-Braking Tuning Effectiveness. In the single-example case (a), the self-braking tuned model exhibits spontaneous termination of overthinking and significantly reduces token usage. On major mathematical benchmarks (b), compared to using OpenR1-Math [1] as the SFT dataset, the self-braking tuned Qwen2.5-Math-1.5B-Instruct [2] achieves a substantial reduction in tokens consumed during inference while maintaining comparable accuracy.

leading to increased computational cost, latency, and redundant reasoning that can obscure core solutions [11]. This "overthinking" [12] poses a significant challenge for practical deployment.

Many recent studies have focused on addressing the problem of overthinking [13, 14], which can be broadly categorized into three approaches: (1) *Model optimization:* Apply reinforcement learning (RL) or supervised fine-tuning (SFT) to equip models with the ability to control reasoning length [15–17]; (2) *Reasoning output optimization*: Dynamically reducing the number of reasoning steps and output length during inference [18–20]; (3) *Adding external restrictions*: Imposing external constraints, such as token budgets, to reduce overthinking [21, 22]. Most existing methods follow the paradigm of external intervention, relying on complex optimization strategies or introducing additional constraint mechanisms, and have yet to fully explore the intrinsic ability of the model to mitigate overthinking on its own.

This reliance on external control prompts a fundamental question: ***Can we enable large reasoning models to autonomously recognize excessive reasoning and terminate their thinking process appropriately?*** Ideally, a model should intrinsically understand when additional reasoning becomes redundant and halt its thought process without external triggers, similar to how humans naturally conclude their reasoning when reaching sufficient certainty.

To address this challenge, we propose **Self-Braking Tuning (SBT)**, a novel framework that teaches LRMs to autonomously identify and terminate redundant reasoning. Unlike previous approaches that impose external constraints, SBT fundamentally reshapes how models perceive and regulate their own reasoning processes. Our key insight is that LRMs can be trained to develop an internal braking mechanism that recognizes when further reasoning becomes unproductive, enabling them to naturally conclude the thought process and transition to formulating the final solution.

Our approach begins with a systematic methodology for identifying overthinking patterns in reasoning trajectories. By combining metrics such as reasoning efficiency ratio and overthinking label ratio, we precisely pinpoint the transition point at which the model shifts from effective reasoning to redundant computation. Based on this analysis, we develop two complementary data construction strategies: (1) Self-Braking Tuning Exact (SBT-E), which strictly removes redundant reasoning segments based on predefined braking points, allowing the model to enter the conclusion phase earlier; (2) Self-Braking Tuning Dynamic (SBT-D), which implements step-level monitoring and dynamically halts reasoning when overthinking patterns emerge.

Building upon the high-quality OpenR1-Math [1] dataset of reasoning trajectories, we constructed two specialized training datasets using these strategies: OpenR1-Math-SBT-E and OpenR1-Math-SBT-D. To further enhance the model's self-awareness of its reasoning state, we introduce braking

prompts at the identified braking points, explicitly simulating the recognition of having sufficiently completed the reasoning process. These prompts enable the model to naturally express an awareness of having reached adequate reasoning depth, thus promoting autonomous termination without the need for external signals. Our experimental results demonstrate consistent improvements in reasoning efficiency across mathematical benchmarks of varying difficulty. While maintaining high accuracy, token consumption was reduced by 30% to 60%.

Our contributions can be summarized as follows:

- We introduce a novel tuning framework that enables LRMs to self-regulate reasoning length without external constraints. Self-Braking Tuning cultivates models' intrinsic ability to recognize and inhibit excessive reasoning, fundamentally improving inference efficiency and response quality.

- We propose a systematic methodology for identifying overthinking patterns and develop two complementary data construction strategies: SBT-E and SBT-D, resulting in specialized training datasets for addressing the overthinking problem. These datasets systematically prune redundant reasoning while preserving essential thinking steps.

- We demonstrate that models trained with our SBT framework maintain original accuracy levels while reducing token consumption by up to 60% across multiple benchmarks, confirming the effectiveness and generalizability of our approach in enhancing reasoning efficiency.

## 2 Related Works

**Large reasoning models** Large reasoning models (LRMs) extend traditional language models with advanced reasoning capabilities, often enabled by reinforcement learning. OpenAI's o1 series [3] marked a key milestone, followed by DeepSeek-R1 [4], which matched o1's performance through a cost-efficient combination of supervised fine-tuning and RL. Subsequent models like Kimi-k1.5 [7] and QwQ-32B [5] further solidified the LRM era. Concurrently, alternative approaches have emerged to reduce reliance on RL, leveraging supervised fine-tuning and data distillation. Models such as DeepSeek-Distill [4], OpenR1-Math-7B [23], Sky-T1 [24], and LIMO [25] have shown that strong reasoning performance can also be achieved without RL, broadening the design space for LRMs. In addition, recent algorithmic advances specifically target robustness and scalability of reasoning [26–28]: $S^3$c-Math [29] introduces spontaneous step-level self-correction to let models detect and fix erroneous intermediate steps during chain-of-thought generation, while InftyThink [30] breaks long-context limits by turning monolithic proofs into iterative short-segment reasoning with concise progress summaries, enabling effectively unbounded reasoning depth with bounded computation.

**Efficient reasoning** Overthinking is a common behavior of large reasoning models, where models generate unnecessarily long answers instead of stopping at the right time. Existing studies addressing overthinking [13] can be grouped into three categories: (1) *Model Optimization:* This line improves model behavior via post-training techniques, primarily reinforcement learning (RL) and supervised fine-tuning (SFT). RL-based methods design length-sensitive rewards to limit output [16, 17, 15, 31]. SFT-based methods use variable-length chain-of-thought (CoT) data and auxiliary constraints to shorten reasoning paths [32–36]. (2) *Reasoning Output Optimization:* This direction reduces reasoning length at inference time by altering generation strategies. *lightthinker* [20] compresses intermediate steps; *DEER* [19] halts once high confidence is reached; *NoThinking* [18] skips reasoning entirely via prompting. These efforts are representative, with many other studies exploring similar directions [37–40]. (3) These methods impose constraints like token budgets or prompt controls to regulate reasoning behavior. In addition to representative works such as Token-Budget [21] and CoD [22], numerous other studies have explored similar constraint-based strategies [41–43].

## 3 Methods

In this section, we begin by analyzing the reasoning trajectories of LRMs to understand the patterns of overthinking (Section 3.1). Based on this analysis, we propose metrics to quantify overthinking (Section 3.2). We then introduce our Self-Braking Tuning framework, which includes two data construction strategies (Section 3.3) and a braking prompt mechanism (Section 3.4).

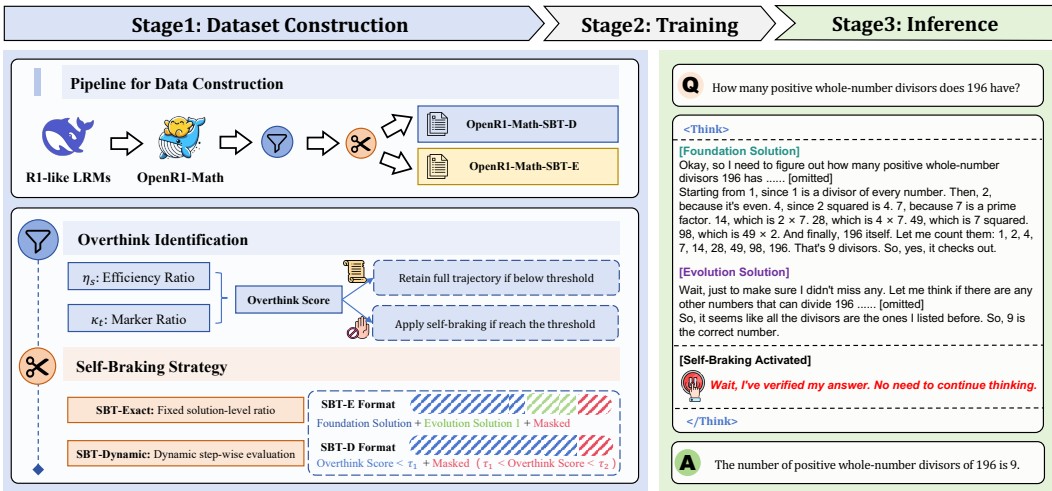

Figure 2: Overview of Self-Braking Tuning. Left: Data construction process with overthinking identification and self braking truncation strategies. Right: An example of automatic reasoning termination in a trained Self-Braking LLM.

## 3.1 Reasoning trajectory analysis in R1-like models

Understanding the reasoning structure of LRMs is key to addressing overthinking. Analysis of trajectories from models like DeepSeek-R1 reveals a common pattern: multiple distinct solution attempts are often generated for a single problem. Based on their role and position, these solution segments can be grouped into two main types:

1. **Foundation solution:** This is the first solution at the beginning of its reasoning process. After comprehending the problem, it proceeds with a step-by-step solution. This forms the foundation of the reasoning process and guides the development of subsequent Evolution Solutions.

2. **Evolution solution:** These solutions appear in the later stages of the model's reasoning process and are often introduced with cues such as "Wait," "Alternatively," or "However." Evolution solutions primarily reflect, refine, supplement, or summarize the foundational solution, and may propose new approaches. While this part of the reasoning grants the model self-correction and improvement capabilities, it is also where overthinking most frequently occurs.

To illustrate how LRMs behave across varying difficulty levels, we constructed a set of math evaluation benchmarks with graduated difficulty and conducted experiments on DeepSeek-Distill-Qwen-7B. The distribution of correct reasoning trajectories is reported in Figure 3. Additionally, a representative example from the MATH500 [44, 45] task is presented to demonstrate the concrete forms of Foundation Solution and Evolution Solution.

## 3.2 Overthinking identification

Identifying overthinking in reasoning trajectories is crucial for designing effective self-braking mechanisms. Based on our structural analysis of reasoning, we propose a quantitative framework for detecting and measuring redundancy in the OpenR1-Math dataset, which spans a range of mathematical problems and difficulty levels. To distinguish essential reasoning from unnecessary computation, we introduce two complementary metrics capturing different forms of redundancy, integrated through a composite scoring mechanism.

**Reasoning efficiency ratio**   Our first metric addresses a fundamental observation in Section 3.1: LRMs frequently derive correct answers relatively early in their reasoning process but continue generating additional solution attempts. To quantify this inefficiency, we introduce the reasoning efficiency ratio, denoted as

$$\eta_s = \frac{FS}{TS} \tag{1}$$

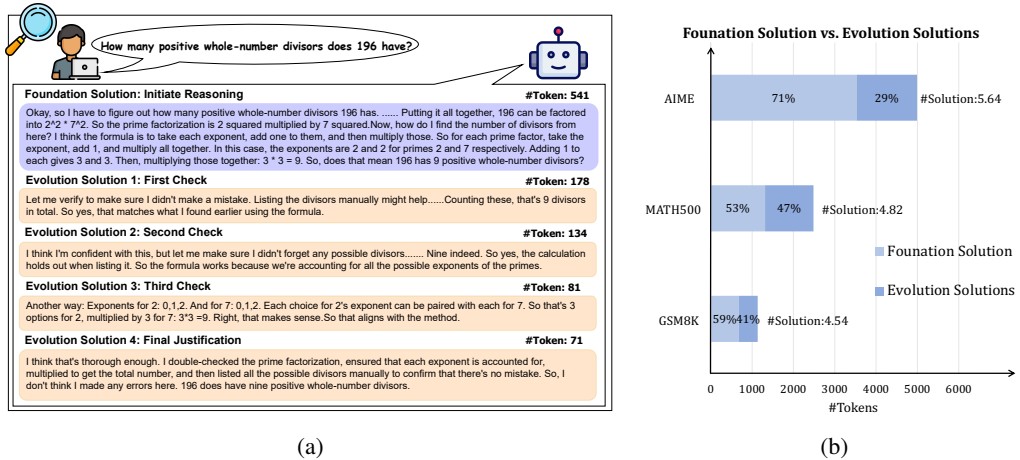

(a)                                              (b)

Figure 3: In panel (a), we present a representative example from DeepSeek-R1-Distill-Qwen-7B. In panel (b), we analyze the model's performance across GSM8K, MATH500, and AIME benchmarks, showing that the Foundation Solution plays a critical role across tasks of varying difficulty.

where $FS$ (first correct steps) represents the number of reasoning steps required to reach the first correct answer within the thinking segment (enclosed by `<think>` and `</think>` tags) and $TS$ (total thinking steps) represents the total number of steps in the entire thinking segment.

This ratio provides a direct measure of reasoning efficiency: values closer to 1 indicate that the model spent most of its reasoning steps before arriving at the correct answer, reflecting a focused and efficient reasoning process. In contrast, values closer to 0 suggest that the model continued reasoning extensively after reaching the correct answer, which may indicate overthinking or redundant computation. The step-based calculation enables us to assess the structural efficiency of the reasoning process independently of implementation-specific token counts.

**Overthinking marker ratio**    While $\eta_s$ captures structural inefficiency, it does not account for linguistic patterns characteristic of overthinking. Through analysis of high-quality DeepSeek-R1 reasoning trajectories, we identified a set of linguistic markers strongly associated with overthinking behaviors—terms that signal reconsideration, verification, or alternative approach exploration. We formalize this observation through the overthinking marker ratio, denoted as $\kappa_t$:

$$\kappa_t = \frac{1}{TT} \sum_{i=1}^{TT} \mathbb{I}[w_i \in \mathcal{M}], \quad \mathbb{I}[w_i \in \mathcal{M}] = \begin{cases} 1, & \text{if } w_i \in \mathcal{M} \\ 0, & \text{otherwise} \end{cases} \tag{2}$$

where $\mathcal{M}$ represents our curated lexicon of overthinking marker terms (complete list in Appendix E), TT (Total Tokens) represents the total number of tokens in the thinking segment, $\mathbb{I}[\cdot]$ is the indicator function that equals 1 when $w_i$ belongs to $\mathcal{M}$ and 0 otherwise.

This ratio quantifies the linguistic footprint of overthinking within a reasoning trajectory. Higher values of $\kappa_t$ indicate a greater presence of reconsideration and verification language, which typically correlates with redundant reasoning patterns.

**Overthink score**    To develop a comprehensive assessment of overthinking that leverages both structural and linguistic indicators, we introduce the overthink score, a weighted combination of our two metrics:

$$\text{Overthink Score} = \beta \times \kappa_t + (1 - \beta) \times (1 - \eta_s) \tag{3}$$

Note that we transform $\eta_s$ to $(1 - \eta_s)$ to ensure directional consistency, as higher values of $\eta_s$ indicate greater efficiency (less overthinking), while higher values of $\kappa_t$ suggest stronger overthinking patterns. The weighting parameter $\beta \in [0, 1]$ balances the contribution of each component to the final score.

In our implementation, we set $\beta = 0.1$ based on both theoretical considerations and empirical validation. This parameter choice reflects two key insights:

- **Reasoning efficiency dominance**: Early arrival at correct answers significantly reduces computational resources and latency, making $(1 - \eta_s)$ the primary component (weighted at 90%). This prioritization aligns with our objective of minimizing unnecessary computation while preserving reasoning quality.

- **Linguistic indicator robustness**: While $\kappa_t$ provides valuable signals about overthinking, it exhibits greater sensitivity to variations in prompt formulation, corpus characteristics, and model-specific output patterns. Assigning a lower weight (10%) to $\kappa_t$ mitigates potential noise amplification from these stylistic fluctuations.

## 3.3 Adaptive inference data construction

Building on our quantitative framework for overthinking identification, we introduce two complementary strategies for constructing adaptive inference length datasets: Self-Braking Tuning Exact (SBT-E) and Self-Braking Tuning Dynamic (SBT-D). Both approaches aim to preserve reasoning depth while cultivating the model's ability to terminate excessive thinking.

**Self-Braking Tuning Exact**  SBT-E implements a solution-level truncation strategy with consistent reasoning structure across all training examples. For each trajectory exhibiting overthinking, we preserve the Foundation Solution plus one Evolution Solution, followed by a small masked segment from subsequent reasoning. This structured approach ensures the model learns clear boundaries between necessary reasoning and excessive computation.

The Foundation Solution captures the initial structured approach to the problem, while the additional Evolution Solution preserves self-correction capabilities. The masked segment, comprising the beginning of the next Evolution Solution, serves as a braking indicator that signals where further reasoning becomes redundant. During training, this masked content does not contribute to the loss function, thereby avoiding reinforcement of overthinking patterns. Algorithm 1 in Appendix presents the formal procedure for SBT-E construction.

**Self-Braking Tuning Dynamic**  While SBT-E uses uniform truncation, SBT-D adopts a step-wise adaptive strategy that tailors reasoning length to each problem. It incrementally analyzes each reasoning step to determine individualized termination points.

The process begins with the full preservation of the Foundation Solution. Subsequent steps are added one by one, with the overthink score recalculated after each. Reasoning continues until the score surpasses a primary threshold $\tau_1$ (set to 0.2), allowing complex problems to retain more steps and simpler ones to terminate earlier.

A masking segment is then defined from steps with overthink scores between $\tau_1$ and a secondary threshold $\tau_2$ (set to $\tau_1 + 5\%$). This segment is excluded from loss computation but retained to expose the model to overthinking patterns without reinforcing them. The full procedure is outlined in Algorithm 2 in Appendix.

Together, SBT-E and SBT-D yield the OpenR1-Math-SBT-E and OpenR1-Math-SBT-D datasets, each with 92,064 examples, designed to train models to autonomously terminate redundant reasoning.

## 3.4 Self-regulating braking strategy

Beyond generating adaptive-length reasoning, we introduce complementary training mechanisms to enhance the model's ability to stop reasoning autonomously. Our Self-Regulating Braking Strategy includes two components: masked redundant thinking and natural language braking signals—both aimed at fostering self-awareness of reasoning efficiency.

**Masked redundant thinking**  While both SBT-E and SBT-D identify optimal truncation points, simply cutting off reasoning there doesn't help models learn to detect overthinking. Instead, we retain a small portion of redundant reasoning and apply loss masking to prevent it from affecting training. For each SBT-E or SBT-D sample, we append this masked segment right after the preserved valid reasoning: in SBT-E, it's the start of the second Evolution Solution; in SBT-D, it includes steps with overthink scores between thresholds $\tau_1$ and $\tau_2$. This consistent strategy across both methods ensures balanced exposure.

By presenting overthinking patterns without calculating their loss, the model learns to distinguish between productive reasoning and redundancy. The masked segment serves as a soft boundary cue, encouraging the model to stop reasoning autonomously during inference.

**Natural language guidance**    We further enhance self-regulation by adding clear natural language cues at reasoning stop points. These braking signals are self-reflective statements that show awareness of finishing reasoning. For example, *"Wait, I've gotten the same answer multiple times, time to end the thinking."*

Placed at the boundary between preserved and masked content, these cues act as linguistic anchors for stopping decisions. Unlike special tokens or external rules, natural language signals fit naturally with the model's abilities, provide explicit metacognitive hints, and keep the reasoning fluent while clearly indicating when to stop.

# 4    Experiments

We conduct extensive experiments to evaluate the effectiveness of Self-Braking Tuning across various model architectures and mathematical reasoning tasks. Our evaluation aims to answer three key questions: (1) How effectively does SBT reduce token consumption while preserving accuracy? (2) How does performance vary across different model sizes and architectures? (3) How do the two SBT variants (SBT-E and SBT-D) compare in practice?

## 4.1    Experimental setup

**Datasets and training.**    We curate a dataset of 92K high-quality instances from OpenR1-Math [1][1] by applying a 16K context limit and filtering out problematic samples (e.g., those with multiple `</think>` tags). This filtered dataset serves as our baseline. We then construct our SBT-E and SBT-D variants following the methodology detailed in Section 3.3.

We perform supervised fine-tuning on both mathematical specialists (Qwen2.5-Math-1.5B/7B-Instruct [2]) and general-purpose models (Llama-3.2-1B and Llama-3.1-8B-Instruct [46]). All models are trained using Megatron-LM for 3 epochs with a 1e-5 initial learning rate, cosine decay schedule, 0.03 warm-up ratio, and 16,384-token maximum sequence length. Training is conducted on 64 Ascend H910B-64G hardware.

**Evaluation benchmarks.**    We evaluate performance across four mathematical reasoning benchmarks of varying difficulty: AIME (24&25, competition-level algebraic problems), AMC23 (pre-collegiate mathematics), MATH500 [44, 45] (diverse mathematical problems), and GSM8K [47] (grade school math word problems). For inference, we use vLLM [48] with temperature 0.7, generating 8 samples per question and reporting Average accuracy. All inference are performed on NVIDIA A100 GPUs.

## 4.2    Main results

Table 1 presents the performance of models trained with our Self-Braking Tuning approaches compared to baselines trained on the unmodified dataset. We observe several significant trends:

**Substantial token reduction with preserved acc.**    Both SBT variants achieve remarkable reductions in token consumption while maintaining comparable accuracy to baseline models. For the Qwen2.5-Math-7B-Instruct model, SBT-E and SBT-D reduce token usage by 30.7% and 23.0% respectively, with accuracy drops of only 2.65% and 1.95%. Even more impressively, when applied to the Llama-3.1-8B-Instruct model, SBT-E reduces token consumption by 62.8% while preserving 94.1% of the baseline accuracy.

**Scaling dynamics across model types**    Efficiency gains from Self-Braking Tuning (SBT) vary by model type. For general-purpose models like Llama, larger models benefit more—token reductions

---

[1]The OpenR1-Math dataset is licensed under the Apache 2.0 License. We adhere to its terms of use and do not redistribute the dataset but build upon it for experimental purposes.

Table 1: Performance of different models with Self-Braking Tuning applied, evaluated across GSM8K, MATH500, AMC23, and AIME (including AIME24 and AIME25) benchmarks.

| Base Model | Method | GSM8K | | MATH500 | | AIME | | AMC23 | | Average | |
|---|---|---|---|---|---|---|---|---|---|---|---|
| | | Acc | #Tok | Acc | #Tok | Acc | #Tok | Acc | #Tok | Acc | #Tok |
| Qwen2.5-Math-1.5B-Instruct | Baseline | 85.00 | 514 | 80.25 | 1712 | 16.25 | 7381 | 55.94 | 3503 | **59.36** | 3277 |
| | SBT-E | 84.85 | 426 | 77.10 | 1121 | 13.75 | 3101 | 55.63 | 2044 | 57.83 | **1673** |
| | SBT-D | 84.87 | 414 | 77.30 | 1046 | 14.17 | 3381 | 50.31 | 1888 | 56.66 | 1682 |
| Qwen2.5-Math-7B-Instruct | Baseline | 96.11 | 1460 | 92.67 | 3816 | 40.83 | 11904 | 83.13 | 6937 | **78.19** | 6029 |
| | SBT-E | 95.45 | 997 | 90.77 | 2501 | 38.75 | 8772 | 77.19 | 4443 | 75.54 | **4178** |
| | SBT-D | 95.37 | 956 | 91.15 | 2629 | 38.38 | 9778 | 80.06 | 5208 | 76.24 | 4643 |
| Llama-3.2-1B-Instruct | Baseline | 41.85 | 1639 | 25.22 | 6624 | 1.25 | 13150 | 9.38 | 10210 | 19.43 | 7906 |
| | SBT-E | 39.96 | 1056 | 24.35 | 3180 | 0.42 | 6615 | 9.06 | 4708 | 18.45 | 3890 |
| | SBT-D | 41.21 | 698 | 25.07 | 2591 | 1.04 | 6821 | 13.13 | 4388 | **20.11** | **3624** |
| Llama-3.1-8B-Instruct | Baseline | 88.03 | 1593 | 59.98 | 9304 | 9.58 | 13663 | 36.75 | 9742 | 48.59 | 8576 |
| | SBT-E | 85.03 | 777 | 57.60 | 2292 | 6.84 | 5658 | 33.44 | 4045 | 45.73 | **3193** |
| | SBT-D | 88.27 | 997 | 62.60 | 3847 | 7.70 | 5845 | 38.12 | 6476 | **49.17** | 4291 |

improve from 54.2% (1B) to 62.8% (8B). In math-specialized models, however, larger models see smaller gains (30.7% for 7B vs. 48.9% for 1.5B), suggesting that specialized models already have more focused and efficient reasoning, leaving less room for further compression. These findings indicate that the self-regulation benefits from SBT depend not only on model scale but also on whether the model is trained for general-purpose or domain-specific tasks.

**SBT-E vs. SBT-D performance.** The two proposed variants show distinct performance characteristics. SBT-E generally achieves greater token reductions (averaging 48.3% across all models compared to 43.9% for SBT-D) but with slightly larger accuracy drops. SBT-D demonstrates more balanced performance, particularly on the most challenging AIME and MATH500 benchmarks. Notably, for the Llama-3.1-8B model, SBT-D actually improves accuracy on MATH500 by 2.62 % while reducing tokens by 58.7%, suggesting that dynamic truncation may help eliminate not just redundant reasoning but potentially harmful overthinking in some cases.

## 5 Analysis

### 5.1 Impact of overthinking thresholds

The threshold for classifying overthinking instances significantly impacts both dataset composition and model performance. We experimented with thresholds of 0.2, 0.3, and 0.4, which classified approximately 60%, 50%, and 40% of samples as overthinking cases, respectively. As shown in Table 2, a threshold of 0.2 yields the best performance for SBT-E, achieving an optimal balance between token reduction (49% fewer tokens than baseline) and accuracy preservation (97.4% of baseline accuracy).

This finding reveals a crucial insight: aggressive overthinking identification (lower thresholds) leads to more substantial efficiency gains without proportional accuracy losses. This suggests that a significant portion of reasoning in LRMs truly is redundant and can be eliminated without compromising problem-solving capabilities. The consistent pattern across both SBT variants indicates that identifying and addressing overthinking in approximately 60% of reasoning instances represents an optimal operating point for Self-Braking Tuning.

### 5.2 Preserved reasoning and redundancy masking trade-off

To develop effective self-braking behavior, models learn both when to continue reasoning and when to stop. We investigated different configurations of preserved (unmasked) and masked content to understand this balance. Shown in Table 3, preserving two complete solutions while masking only a few additional sentences yields optimal performance, reducing tokens by 49% while maintaining 97.4% of baseline accuracy.

Table 2: Performance across overthink score thresholds. Detailed results shown in Appendix Table 12.

| Method | Threshold | Acc | #Tok |
|---|---|---|---|
| Baseline | – | 59.36 | 3278 |
| SBT-Exact | 0.2 | **57.83** | **1673** |
| | 0.3 | 56.70 | 1755 |
| | 0.4 | 57.38 | 1834 |
| SBT-Dynamic | 0.2 | 56.66 | **1682** |
| | 0.3 | **57.47** | 1917 |
| | 0.4 | 57.36 | 1902 |

Table 3: Performance with different configurations of preserved reasoning and masked redundant content. Detailed results shown in Appendix Table 13.

| Reservations & Masked Content | Acc | #Tok |
|---|---|---|
| Baseline | 59.36 | 3277 |
| 1 solution & A few sentences | 56.95 | 1700 |
| 1 solution & 1 solution | 57.69 | 1697 |
| 2 solutions & A few sentences | **57.83** | **1673** |
| 2 solutions & 1 solution | 57.45 | 1684 |

This finding provides two key insights. First, solution repetition serves as a natural termination signal: when a model derives the same answer twice, it learns this is a strong indication to conclude reasoning. Second, we observe an inverse relationship between preserved and masked content: with more preserved reasoning (two solutions), less masked content is optimal; with less preserved reasoning (one solution), more masked content performs better.

This relationship suggests that models require a certain "reasoning quota" to develop robust problem-solving capabilities, which can be satisfied either through more preserved reasoning or more exposure to masked reasoning patterns. However, the superior performance of the "two solutions with minimal masked content" configuration indicates that clearly delineated, complete reasoning paths provide stronger learning signals than exposure to additional masked content.

## 5.3 With vs. without masked redundant thinking

To assess masked redundant thinking (MRT), we compare SBT-E with and without masked segments. In the ablation variant, trajectories truncate directly without retaining masked content.

Results show removing MRT yields minimal accuracy gain (+0.19%) but substantial efficiency loss (+37.8% tokens). Without exposure to masked overthinking patterns, models fail to internalize what constitutes redundancy. Direct truncation provides only implicit stopping signals—models see where reasoning ends but not why. Masked segments serve as explicit negative examples: models observe overthinking without gradient reinforcement, enabling discriminative learning of termination boundaries. This exposure-without-reinforcement principle proves essential for self-regulation while preserving reasoning quality.

Table 4: MRT comparison. Detailed results shown in Appendix Table 14.

| Config. | Acc | #Tok |
|---|---|---|
| Baseline | 59.36 | 3277 |
| w/ MRT | **57.83** | **1673** |
| w/o MRT | 58.02 | 2306 |

## 5.4 Impact of $\beta$ on RER-OMR balance

The overthink score (Equation 3) combines structural efficiency $(1 - \eta_s)$ and linguistic markers $\kappa_t$ through weighting parameter $\beta$. Table 5 examines sensitivity across $\beta \in \{0.05, 0.1, 0.15, 0.2\}$.

Results show $\beta = 0.1$ consistently achieves optimal performance for both SBT variants, with 57.83% accuracy (SBT-E) and 56.66% (SBT-D) while maintaining strong efficiency (1673 and 1682 tokens). Lower values ($\beta = 0.05$) over-emphasize linguistic markers, degrading accuracy by 0.4–1.4 points despite minimal token savings. Higher values ($\beta \geq 0.15$) reduce linguistic contribution, causing accuracy drops of 1.3–2.0 points with increased token consumption. This validates prioritizing structural efficiency (90%) while retaining linguistic signals (10%) for robust detection without prompt sensitivity.

Table 5: Performance with different $\beta$ values for overthink score weighting. Detailed results shown in Appendix Table 15.

| Method | $\beta$ | Acc | #Tok |
|---|---|---|---|
| Baseline | – | 59.36 | 3277 |
| SBT-E | 0.05 | 56.48 | 1762 |
| | **0.1** | **57.83** | **1673** |
| | 0.15 | 56.52 | 1874 |
| | 0.2 | 55.86 | 1809 |
| SBT-D | 0.05 | 56.24 | **1678** |
| | **0.1** | **56.66** | 1682 |
| | 0.15 | 56.21 | 1784 |
| | 0.2 | 55.74 | 1814 |

## 5.5 Step-level vs. token-level overthinking detection

The granularity of overthinking detection, whether operating at the reasoning step level or token level, impacts both the coherence of preserved reasoning and overall model performance. We compared our step-level approach with a token-level alternative using a token efficiency ratio defined as $\eta_t = \frac{FT}{TT}$, where $FT$ represents tokens until first correct answer and $TT$ represents total tokens.

Results in Table 6 demonstrate that step-level detection outperforms token-level approaches, achieving both higher accuracy and lower token usage. This confirms our hypothesis that reasoning coherence is better preserved when entire logical steps are maintained intact. Token-level truncation, while more granular, risks breaking logical units of reasoning, potentially creating disjointed or incomplete thinking patterns that are harder for models to learn from or reproduce effectively.

This finding highlights the importance of respecting the inherent structure of reasoning when developing overthinking mitigation strategies: models benefit from complete logical units rather than more aggressive but potentially incoherent truncation approaches.

Table 6: Overthinking detection granularity comparison. Detailed results in Appendix Table 16.

| Level | Acc | #Tok |
|---|---|---|
| Baseline | 59.36 | 3277 |
| Step-Level | **56.66** | **1682** |
| Token-Level | 56.24 | 1753 |

Table 7: Guiding mode comparison. Detailed results shown in Appendix Table 17.

| Guiding Mode | Acc | #Tok |
|---|---|---|
| Baseline | 59.36 | 3277 |
| Natural Language | **56.66** | **1682** |
| Special Token | 56.61 | 1797 |
| No Guidance | 56.39 | 1801 |

## 5.6 Natural language guidance vs. alternative approaches

A fundamental aspect of Self-Braking Tuning is the mechanism used to signal reasoning termination. We compared our natural language guidance approach (epiphany sentences like "I've verified my answer, no need to continue...") with both a special token approach using `<stop_overthinking>` and a configuration without any explicit guidance.

Table 7 shows natural language guidance achieves optimal performance (56.66% accuracy, 1682 tokens). The necessity of explicit termination signals is confirmed by the no-guidance ablation, which degrades both accuracy ($-0.27\%$) and efficiency ($+7.1\%$ tokens). Compared to special tokens (56.61%, 1797 tokens), natural language guidance maintains comparable accuracy with 6.4% fewer tokens by leveraging models' existing semantic understanding of logical transitions rather than requiring learning of artificial control conventions.

## 6 Conclusion

In this paper, we propose a novel endogenous approach, Self-Braking Tuning (SBT), to mitigating overthinking in large language models. SBT aims to stimulate the model's ability to autonomously identify and stop redundant reasoning. We construct a data framework with adaptive reasoning lengths, where overthinking characteristics are extracted through step-level analysis and keyword annotation. Based on this, we design two data generation strategies, SBT-E and SBT-D, to help the model learn when to stop and how to simplify its reasoning process. During supervised fine-tuning, we introduce redundancy masking and epiphany sentences to preserve the reasoning paradigm while enhancing the model's sensitivity to redundant thought patterns. Experimental results show that SBT models significantly reduce token consumption by 30%–60% on multiple mathematical benchmarks, with minimal impact on accuracy. Our work demonstrates the potential for large reasoning models to self-regulate their reasoning and offers a promising direction for more efficient long-chain reasoning.

## Acknowledgement

This work is supported by the National Natural Science Foundation of China (No. 62376245), National Key Research and Development Project (No. 2024YFB3312900), the Key Research and Development Program of Zhejiang Province, China (No. 2024C03255), the Fundamental Research Funds for the Central Universities (226-2024-00170), MOE Engineering Research Center of Digital Library, CCF-Tencent Rhino-Bird Open Research Fund, and ZJU Kunpeng&Ascend Center of Excellence.

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

# A  Limitations

While *Self-Braking Tuning* (SBT) has demonstrated its ability to reduce overthinking by enabling models to autonomously regulate reasoning length, several limitations remain:

- **Domain generalization.** Our study focuses primarily on math reasoning tasks (e.g., GSM8K, MATH), which, while structurally rich and sensitive to overthinking, do not cover the full diversity of reasoning challenges. The applicability of SBT to open-ended, commonsense, logical, or multimodal reasoning remains unverified. These domains may exhibit distinct redundancy patterns that require specialized adaptation strategies.

- **Data scalability and adaptivity.** Due to computational constraints, our experiments are conducted on a dataset of approximately 92K examples. The impact of scaling to larger datasets (e.g., millions of samples) remains unexplored. Moreover, current data construction strategies (SBT-E/D) rely on fixed threshold parameters for overthinking detection, which may require manual tuning across different tasks and hinder dynamic adaptation.

- **Overthinking signal definition.** We use reasoning efficiency ($\eta_s$) and overthink marker ratio ($\kappa_t$) to estimate redundancy, but these metrics may not capture all subtle or latent forms of reasoning utility. Some steps that appear redundant may actually serve a hidden purpose, such as helping the model organize its reasoning or implicitly represent abstract patterns.

- **Interpretability and controllability.** While SBT introduces a soft constraint via loss masking, the internal process by which a model decides to terminate reasoning remains opaque. There is no explicit mechanism to trace or control this decision, limiting transparency and reliability—especially in applications requiring high interpretability, such as education, healthcare, or scientific domains.

- **Potential trade-offs in complex tasks.** For tasks requiring deep, multi-step reasoning (e.g., theorem proving), prematurely terminating reasoning may risk omitting critical steps. Although SBT preserves accuracy in most cases, it may underperform in scenarios where full-chain reasoning is essential. Adaptive or progressive braking strategies may be needed to better balance efficiency and completeness.

Future work could address these limitations by (i) expanding SBT to open-ended and multimodal tasks, (ii) scaling to larger and more diverse datasets, (iii) developing adaptive thresholding and automatic data construction pipelines, (iv) improving multilingual and domain-robust prompt designs, and (v) integrating interpretable or feedback-driven stopping mechanisms that better align with complex reasoning dynamics.

# B  Autonomous reasoning regulation in self-braking models

A critical distinction of our Self-Braking Tuning framework is that SBT-E and SBT-D are *data construction strategies*, not inference-time control mechanisms. After training, models exhibit autonomous reasoning regulation without external intervention, dynamically adapting reasoning depth to problem complexity.

## B.1  Emergent self-regulation behavior

To validate that self-braking emerges naturally during inference, we analyzed reasoning patterns of Qwen2.5-Math-7B-Instruct-SBT-E on AIME. Models spontaneously generate epiphany sentences (e.g., "Wait, I've verified my answer. No need to continue thinking.") and terminate without external signals.

Table 8: Analysis of autonomous early termination in SBT-trained models on AIME. Models naturally terminate reasoning in about 50% of cases, achieving both higher accuracy and 48–51% token reduction.

| Dataset | Exit Type | % of Cases | Acc | #Tok |
|---------|-----------|------------|-----|------|
| AIME24 | Early Exit | 50.83% | 41.80% | 5,692 |
|        | No Early Exit | 49.17% | 38.98% | 11,084 |
| AIME25 | Early Exit | 49.17% | 41.53% | 6,483 |
|        | No Early Exit | 50.83% | 32.79% | 12,201 |

Table 8 shows that self-braking is not enforced through hard constraints. Cases with early termination achieve higher accuracy while using fewer tokens—indicating models learn to recognize when additional reasoning becomes counterproductive.

## B.2 Problem-adaptive reasoning depth

SBT-trained models autonomously adjust reasoning length based on task difficulty:

Table 9: Adaptive reasoning depth of Qwen2.5-Math-7B-Instruct-SBT-E across difficulty levels. The 7.3× variation between GSM8K and AIME25 demonstrates autonomous complexity adaptation without external difficulty indicators.

| Dataset | Difficulty | Avg Steps |
|---------|-----------|-----------|
| GSM8K | Easy | 27.78 |
| MATH500 | Medium | 51.32 |
| AMC23 | Hard | 106.40 |
| AIME25 | Very Hard | 202.23 |

The substantial variation (27.78 to 202.23 steps) occurs without external difficulty indicators or task-specific prompting, suggesting internalized problem complexity assessment.

## B.3 Interpretation of self-regulation mechanisms

The training process exposes models to three key signals: (1) preserved reasoning demonstrating sufficient problem-solving depth, (2) natural language cues marking reasoning completion, and (3) masked redundant segments that models observe but do not learn to reproduce. This combination appears to develop an internal representation of "reasoning sufficiency" that activates during generation.

We hypothesize that the natural language braking prompts (e.g., "Wait, I've gotten the same answer multiple times") serve as anchor points in the representation space, allowing models to recognize similar reasoning states during inference and trigger corresponding termination behavior. The consistency of early termination rates (∼50% on AIME) across individual problems suggests this behavior generalizes beyond memorization of training examples.

# C Out-of-domain generalization

To evaluate whether Self-Braking Tuning generalizes beyond mathematical reasoning, we test SBT-trained models on two out-of-domain benchmarks covering diverse knowledge domains: MMLU-Redux (general knowledge and reasoning) [49] and GPQA-Diamond (graduate-level science questions) [50]. All models were trained exclusively on math data (OpenR1-Math), with no domain-specific fine-tuning for these benchmarks.

Table 10: Out-of-domain evaluation on MMLU-Redux and GPQA-Diamond. Models trained exclusively on mathematical reasoning data demonstrate consistent efficiency gains (26–65% token reduction) across non-mathematical tasks, with accuracy drops typically limited to 1–3%.

| Base Model | Method | MMLU-Redux | | GPQA-Diamond | | Average | |
|------------|--------|------|------|------|------|------|------|
| | | Acc | #Tok | Acc | #Tok | Acc | #Tok |
| Qwen2.5-Math-1.5B | Baseline | 45.84 | 2,061 | 24.75 | 5,485 | 35.30 | 3,773 |
| | SBT-E | 43.12 | **1,403** | 25.06 | 3,194 | 34.09 | 2,299 |
| | SBT-D | **43.28** | 1,566 | **26.20** | **3,002** | **34.74** | **2,284** |
| Qwen2.5-Math-7B | Baseline | 67.04 | 3,229 | 41.15 | 8,892 | 54.10 | 6,061 |
| | SBT-E | 65.84 | **1,927** | 40.40 | **6,205** | 53.12 | **4,066** |
| | SBT-D | **66.39** | 1,998 | **41.29** | 6,706 | **53.84** | 4,352 |
| Llama-3.2-1B | Baseline | 35.62 | 1,933 | 17.99 | 9,321 | 26.81 | 5,627 |
| | SBT-E | 32.24 | 770 | **24.24** | 3,516 | 28.24 | 2,143 |
| | SBT-D | **33.12** | **725** | 23.48 | **3,157** | **28.30** | **1,941** |
| Llama-3.1-8B | Baseline | 80.53 | 2,481 | 37.31 | 8,918 | 58.92 | 5,699 |
| | SBT-E | 77.46 | **1,646** | 36.30 | **5,346** | 56.88 | **3,496** |
| | SBT-D | **77.74** | 1,668 | **36.91** | 6,717 | **57.33** | 4,192 |

**Cross-domain transferability.** SBT achieves 26–65% token reduction across non-mathematical benchmarks despite training exclusively on math data, with minimal accuracy degradation (typically 1–3%). This demonstrates that self-regulation learned from mathematical reasoning generalizes to diverse knowledge domains.

**Model size effects.** Efficiency gains scale inversely with model size: smaller models (Llama-3.2-1B) achieve 65% token reduction, while larger models (Llama-3.1-8B, Qwen2.5-Math-7B) show 26–40% reductions. Notably, Llama-3.2-1B with SBT-E improves GPQA-Diamond accuracy by 6.25 percentage points ($17.99\% \rightarrow 24.24\%$), indicating that overthinking mitigation can enhance performance on challenging out-of-domain tasks for smaller models.

**Architecture consistency.** Both domain-specialized (Qwen2.5-Math) and general-purpose (Llama) models benefit comparably from SBT on out-of-domain evaluation, confirming that self-braking represents a transferable meta-cognitive capability rather than task-specific adaptation.

These results establish that reasoning efficiency patterns learned from mathematics transfer broadly to general knowledge and scientific reasoning, supporting SBT's applicability beyond its training domain.

# D  Extended threshold analysis

To comprehensively evaluate the robustness of our overthinking detection mechanism, we conducted an extended threshold analysis spanning $\tau_1 \in \{0.05, 0.1, 0.2, 0.3, 0.4, 0.5\}$. This analysis examines how threshold selection affects both the proportion of samples classified as overthinking and the resulting model performance.

Table 11: Extended threshold analysis for overthinking score across $\tau_1 \in [0.05, 0.5]$. Results Averaged across GSM8K, MATH500, AIME, and AMC23. Overthink % indicates the proportion of samples classified as containing overthinking at each threshold.

| Method | Threshold | Overthink % | Accuracy | #Tokens | Reduction |
|---|---|---|---|---|---|
| Baseline | – | – | 59.36 | 3,277 | – |
| SBT-E | 0.05 | 75.20 | 55.14 | 1,407 | 57.1% |
| | 0.1 | 65.40 | 55.59 | 1,427 | 56.5% |
| | **0.2** | **60.30** | **57.83** | **1,673** | **48.9%** |
| | 0.3 | 50.20 | 56.70 | 1,755 | 46.4% |
| | 0.4 | 41.00 | 57.38 | 1,834 | 44.0% |
| | 0.5 | 2.06 | 57.12 | 2,602 | 20.6% |
| SBT-D | 0.05 | 74.20 | 54.90 | 1,215 | 62.9% |
| | 0.1 | 62.30 | 55.90 | 1,251 | 61.8% |
| | **0.2** | **62.50** | **56.66** | **1,682** | **48.7%** |
| | 0.3 | 50.90 | 57.47 | 1,917 | 41.5% |
| | 0.4 | 40.10 | 57.36 | 1,902 | 42.0% |
| | 0.5 | 0.19 | 57.09 | 2,696 | 17.7% |

## D.1  Threshold regime characterization

The extended analysis reveals three distinct performance regimes:

**Aggressive pruning ($\tau_1 < 0.2$).** Low thresholds classify 62–75% of samples as overthinking, achieving maximum token reduction (57–63%) but incurring significant accuracy degradation (2.5–4.5 percentage points). At $\tau_1 = 0.05$, the framework removes reasoning content too aggressively, truncating even necessary exploration steps in complex problems. This regime prioritizes efficiency over reasoning completeness.

**Balanced operation ($\tau_1 \in [0.2, 0.4]$).** This range identifies 40–62% of samples as overthinking and demonstrates stable performance with <1.2% accuracy variation while maintaining 41–49% token reduction. The relative insensitivity to exact threshold values within this range indicates robust overthinking detection. Peak performance occurs at $\tau_1 = 0.2$ for both SBT variants, achieving optimal accuracy-efficiency trade-offs (57.83% / 48.9% reduction for SBT-E; 56.66% / 48.7% for SBT-D).

**Conservative pruning ($\tau_1 = 0.5$).** High thresholds classify only 0.2–2% of samples as overthinking, causing the framework to degenerate toward baseline behavior. Token reduction drops to 18–21% with no compensatory accuracy gains, indicating that insufficient overthinking mitigation fails to realize efficiency benefits.

## D.2  Threshold selection rationale

We select $\tau_1 = 0.2$ as the default threshold based on three considerations:

**Performance optimality.** This threshold achieves the best accuracy within the balanced regime while maintaining substantial efficiency gains across both SBT variants and all evaluated benchmarks.

**Empirical stability.** The plateau in the $[0.2, 0.4]$ range demonstrates that performance degrades gradually rather than catastrophically under threshold perturbations, supporting practical deployability.

**Dataset composition.** A threshold of 0.2 creates approximately 60% SBT-processed and 40% original trajectories, enabling models to learn both when to terminate (from processed examples) and when to continue reasoning (from preserved examples). This balanced exposure prevents over-generalization of braking behavior while establishing clear termination patterns.

These findings establish that SBT's effectiveness stems from systematic overthinking identification rather than aggressive truncation, with the optimal operating point balancing reasoning preservation against redundancy elimination.

# E    Overthink markers

In the study of overthinking behaviors, several prior works have highlighted that certain words associated with reflection, hesitation, or backtracking play a critical role in identifying and guiding redundant reasoning processes [17, 51–53]. Building on these insights, we compile a set of common Overthink Markers, including:

| | | |
|---|---|---|
| Another | Backtrack | But |
| Check | Going back | Hmm |
| Hmmm | However | Hold on |
| Instead of | Just to be thorough | Just to make sure |
| Let me check | Let me just double-check | Let me try another |
| Let me verify | Maybe | Maybe I can consider |
| Maybe I should consider | Might | Not sure |
| Perhaps | Recheck | Retry |
| Trace back | Wait | |

These terms frequently co-occur with behaviors such as repeated verification, alternative hypothesis formulation, or reasoning path retracing, and are therefore treated as linguistic indicators of redundant cognitive load during reasoning. Based on this, we construct the set $\mathcal{M}$ as part of our overthinking detection metric, used to compute redundancy density at the linguistic level and help identify potential efficiency bottlenecks in deep reasoning.

# F    Algorithms of self-braking tuning

To facilitate reproducibility and offer a transparent view into our data construction pipeline, we provide the formal procedures for the Self-Braking Tuning Exact (SBT-E) and Self-Braking Tuning Dynamic (SBT-D) methods in this appendix. These two strategies represent complementary approaches for curating training data that teach models to regulate their own reasoning length and avoid excessive computation.

SBT-E adopts a uniform truncation scheme, where each reasoning trajectory is truncated at a consistent structural boundary: the Foundation Solution and the first Evolution Solution are preserved, followed by a small masked portion of subsequent reasoning. This approach provides a clean and interpretable signal for identifying the onset of overthinking.

In contrast, SBT-D offers a fine-grained, adaptive strategy that dynamically determines the optimal stopping point for each problem based on the model's own overthinking scores. It incrementally evaluates reasoning steps, retaining those below a primary overthinking threshold, and masks additional steps that exceed this threshold but remain below a secondary cutoff—effectively preserving problem-specific nuances in reasoning depth.

Crucially, both methods employ loss masking on the redundant segments: the model sees these overthinking patterns during training but receives no gradient updates from them. This enables the model to implicitly recognize and learn to avoid overthinking, without being rewarded for producing verbose or unnecessary steps.

The full algorithmic details are outlined in Algorithms 1 and 2, which serve as a blueprint for implementing the Self-Braking Tuning framework.

---

**Algorithm 1** Self-Braking Tuning Exact (SBT-E)

---

**Require:** Reasoning trajectory $T$ with Foundation Solution $FS$ and Evolution Solutions $ES = [ES_1, ES_2, ...]$
**Ensure:** Modified trajectory $T'$ with preserved and masked segments
  1: PreservedSegment $\leftarrow FS + ES_1$
  2: **if** $|ES| > 1$ **then**
  3:     MaskedSegment $\leftarrow$ First 10-20% of $ES_2$
  4: **else**
  5:     MaskedSegment $\leftarrow \emptyset$
  6: **end if**
  7: $T' \leftarrow$ PreservedSegment + MaskedSegment {With loss masking on MaskedSegment}
  8: **return** $T'$

---

---

**Algorithm 2** Self-Braking Tuning Dynamic (SBT-D)

---

**Require:** Reasoning trajectory $T$ with steps $[S_1, S_2, ..., S_n]$, thresholds $\tau_1$ and $\tau_2$
**Ensure:** Modified trajectory $T'$ with preserved and masked segments
  1: PreservedThinking $\leftarrow$ Foundation Solution from $T$
  2: $i \leftarrow$ index of first step after Foundation Solution
  3: **while** $i \leq n$ **and** CalculateOverthinkScore(PreservedThinking + $S_i$) < $\tau_1$ **do**
  4:     PreservedThinking $\leftarrow$ PreservedThinking + $S_i$
  5:     $i \leftarrow i + 1$
  6: **end while**
  7: MaskedThinking $\leftarrow \emptyset$
  8: **while** $i \leq n$ **and** CalculateOverthinkScore(PreservedThinking + MaskedThinking + $S_i$) < $\tau_2$ **do**
  9:     MaskedThinking $\leftarrow$ MaskedThinking + $S_i$
 10:     $i \leftarrow i + 1$
 11: **end while**
 12: $T' \leftarrow$ PreservedThinking + MaskedThinking {With loss masking on MaskedThinking}
 13: **return** $T'$

---

# G   Detailed experimental results

In Section 5, to improve the readability of the main text, we only present the Average results across the datasets. Here, we provide the specific data and evaluation results.

Table 12: Performance with varying overthink score thresholds. Lower thresholds (0.2) achieve optimal efficiency-accuracy trade-offs—up to 49% token reduction while preserving 97.4% baseline accuracy—validating aggressive pruning effectiveness. Accuracy remains stable across thresholds on simple tasks (GSM8K), while complex tasks (AIME) show marginal accuracy gains at higher thresholds (0.3–0.4) with increased token costs, indicating task-dependent sensitivity to pruning aggressiveness.

| Method | Threshold | GSM8K | | MATH500 | | AIME | | AMC23 | | Average | |
|---|---|---|---|---|---|---|---|---|---|---|---|
| | | Acc | #Tok | Acc | #Tok | Acc | #Tok | Acc | #Tok | Acc | #Tok |
| Baseline | - | 85.00 | 514 | 80.25 | 1712 | 16.25 | 7381 | 55.94 | 3503 | 59.36 | 3277 |
| SBT-Exact | 0.2 | 84.85 | 426 | 77.10 | 1121 | 13.75 | 3101 | 55.63 | 2044 | **57.83** | **1673** |
| | 0.3 | 85.16 | 424 | 77.25 | 1113 | 15.63 | 3353 | 48.75 | 2132 | 56.70 | 1755 |
| | 0.4 | 84.73 | 421 | 77.40 | 1130 | 12.71 | 3795 | 54.69 | 1988 | 57.38 | 1834 |
| SBT-Dynamic | 0.2 | 84.87 | 414 | 77.30 | 1046 | 14.17 | 3381 | 50.31 | 1888 | 56.66 | **1682** |
| | 0.3 | 84.58 | 410 | 78.00 | 1125 | 12.92 | 3710 | 54.37 | 2422 | **57.47** | 1917 |
| | 0.4 | 85.07 | 407 | 78.73 | 1187 | 14.38 | 3593 | 51.25 | 2421 | 57.36 | 1902 |

Table 13: Performance corresponding to different combinations of preserved and masked reasoning. The best configuration—preserving two complete solutions while masking only a few redundant sentences—achieves the highest Average accuracy (57.83%) with 49% fewer tokens than baseline. On simpler datasets like GSM8K, even minimal preservation suffices for learning effective termination, while harder tasks such as AIME benefit more from exposing multiple complete solutions, highlighting that task complexity influences the optimal balance between reasoning exposure and truncation cues.

| Reservations & Masked Content | GSM8K | | MATH500 | | AIME | | AMC23 | | Average | |
|---|---|---|---|---|---|---|---|---|---|---|
| | Acc | #Tok | Acc | #Tok | Acc | #Tok | Acc | #Tok | Acc | #Tok |
| Baseline | 85.00 | 514 | 80.25 | 1712 | 16.25 | 7381 | 55.94 | 3503 | 59.36 | 3277 |
| 1 solution & A few sentences | 85.23 | 416 | 78.00 | 1103 | 12.71 | 3132 | 51.88 | 2148 | 56.95 | 1700 |
| 1 solution & 1 solution | 85.06 | 432 | 78.60 | 1101 | 13.96 | 3178 | 53.12 | 2076 | 57.69 | 1697 |
| 2 solutions & A few sentences | 84.85 | 426 | 77.10 | 1121 | 13.75 | 3101 | 55.63 | 2044 | **57.83** | **1673** |
| 2 solutions & 1 solution | 84.77 | 411 | 77.82 | 1034 | 12.50 | 3092 | 54.69 | 2197 | 57.45 | 1684 |

Table 14: Performance comparison of Masked Redundant Thinking mechanism across benchmarks of varying difficulty. Removing MRT yields marginal Average accuracy gain (+0.19%) but substantial token increase (+37.8%), with dataset-dependent effects: simple tasks (GSM8K: +9.6% tokens) show minimal impact, while complex reasoning tasks experience severe efficiency degradation (AIME: +46.5% tokens, AMC: +36.4% tokens). These results confirm MRT enables efficient termination learning through exposure-without-reinforcement, with benefits scaling with task complexity.

| Configuration | GSM8K | | MATH500 | | AIME | | AMC | | Average | |
|---|---|---|---|---|---|---|---|---|---|---|
| | Acc | #Tok | Acc | #Tok | Acc | #Tok | Acc | #Tok | Acc | #Tok |
| Baseline | 85.00 | 514 | 80.25 | 1712 | 16.25 | 7381 | 55.94 | 3503 | 59.36 | 3277 |
| w/ MRT | **84.85** | **426** | **77.10** | **1121** | **13.75** | **3101** | **55.63** | **2044** | **57.83** | **1673** |
| w/o MRT | 85.05 | 467 | 78.30 | 1425 | 14.38 | 4544 | 54.37 | 2788 | 58.02 | 2306 |

Table 15: Performance with different $\beta$ values for overthink score weighting across benchmarks. While $\beta = 0.1$ achieves best Average performance for both variants (57.83% SBT-E, 56.66% SBT-D), optimal values vary by dataset: $\beta = 0.05$ maximizes accuracy on simple tasks (GSM8K) and minimizes tokens on MATH500, while $\beta = 0.1$ dominates on complex tasks (AIME, AMC23). Lower $\beta$ over-emphasizes linguistic markers with limited efficiency gains, while higher $\beta$ degrades both accuracy and efficiency. This validates $\beta = 0.1$ as the robust default, balancing structural efficiency (90%) and linguistic signals (10%) across task complexities.

| Method | $\beta$ | GSM8K | | MATH500 | | AIME | | AMC23 | | Average | |
|---|---|---|---|---|---|---|---|---|---|---|---|
| | | Acc | #Tok | Acc | #Tok | Acc | #Tok | Acc | #Tok | Acc | #Tok |
| Baseline | – | 85.00 | 514 | 80.25 | 1712 | 16.25 | 7381 | 55.94 | 3503 | 59.36 | 3277 |
| SBT-E | 0.05 | **84.92** | 433 | 76.45 | **1089** | 13.13 | 3198 | 51.42 | 2328 | 56.48 | 1762 |
| | 0.1 | 84.85 | **426** | **77.10** | 1121 | **13.75** | **3101** | **55.63** | **2044** | **57.83** | **1673** |
| | 0.15 | 84.78 | 448 | 76.80 | 1243 | 13.54 | 3487 | 50.96 | 2318 | 56.52 | 1874 |
| | 0.2 | 84.61 | 441 | 76.05 | 1198 | 12.92 | 3421 | 49.86 | 2176 | 55.86 | 1809 |
| SBT-D | 0.05 | **84.89** | 421 | 76.50 | 1087 | 13.33 | **3289** | 50.24 | 1915 | 56.24 | **1678** |
| | 0.1 | 84.87 | **414** | **77.30** | **1046** | **14.17** | 3381 | **50.31** | **1888** | **56.66** | 1682 |
| | 0.15 | 84.83 | 437 | 76.95 | 1178 | 13.75 | 3598 | 49.31 | 1923 | 56.21 | 1784 |
| | 0.2 | 84.71 | 429 | 76.30 | 1154 | 13.13 | 3701 | 48.82 | 1972 | 55.74 | 1814 |

Table 16: Performance comparison between step-level and token-level overthinking detection. Step-level supervision consistently achieves lower token usage across all datasets (e.g., 414 vs. 431 on GSM8K, 1888 vs. 2091 on AMC23), indicating more efficient reasoning truncation. Accuracy-wise, the two methods are comparable on GSM8K and MATH500, but step-level clearly outperforms token-level on more challenging tasks such as AIME (14.17% vs. 11.34%). These results highlight that preserving complete reasoning steps enables better efficiency–accuracy trade-offs, especially for complex problem-solving.

| Level | GSM8K | | MATH500 | | AIME | | AMC23 | | Average | |
|---|---|---|---|---|---|---|---|---|---|---|
| | Acc | #Tok | Acc | #Tok | Acc | #Tok | Acc | #Tok | Acc | #Tok |
| Baseline | 85.00 | 514 | 80.25 | 1712 | 16.25 | 7381 | 55.94 | 3503 | 59.36 | 3277 |
| Step-Level | 84.87 | 414 | 77.30 | 1046 | 14.17 | 3381 | 50.31 | 1888 | **56.66** | **1682** |
| Token-Level | 85.09 | 431 | 78.23 | 1088 | 11.34 | 3399 | 50.31 | 2091 | 56.24 | 1753 |

Table 17: Performance comparison of different guiding mechanisms for reasoning termination. Natural language guidance consistently achieves the best efficiency-accuracy trade-off across benchmarks, with 3381 vs. 3647 tokens on AIME and 1888 vs. 2007 on AMC compared to special tokens. The ablation without any guidance mechanism (No Guidance) demonstrates the necessity of explicit termination signals, showing degraded performance with both reduced accuracy ($-0.27\%$) and increased token consumption ($+7.1\%$) compared to natural language guidance. These results confirm that semantically aligned, self-reflective cues enable more effective reasoning regulation than explicit control tokens or implicit learning alone.

| Guiding Mode | GSM8K | | MATH500 | | AIME | | AMC | | Average | |
|---|---|---|---|---|---|---|---|---|---|---|
| | Acc | #Tok | Acc | #Tok | Acc | #Tok | Acc | #Tok | Acc | #Tok |
| Baseline | 85.00 | 514 | 80.25 | 1712 | 16.25 | 7381 | 55.94 | 3503 | 59.36 | 3277 |
| Natural Language | **84.87** | **414** | **77.30** | **1046** | **14.17** | **3381** | 50.31 | **1888** | **56.66** | **1682** |
| Special Token | 84.94 | 413 | 77.92 | 1120 | 12.34 | 3647 | **51.25** | 2007 | 56.61 | 1797 |
| No Guidance | 84.73 | 421 | 77.05 | 1098 | 13.54 | 3523 | 50.24 | 2162 | 56.39 | 1801 |

# NeurIPS Paper Checklist

1. **Claims**

   Question: Do the main claims made in the abstract and introduction accurately reflect the paper's contributions and scope?

   Answer: [Yes]

   Justification: The abstract and introduction clearly state the main contributions of the paper, which include proposing a novel framework called Self-Braking Tuning (SBT) to address overthinking in large reasoning models. They highlight the methodology for identifying overthinking patterns, constructing adaptive-length reasoning datasets (SBT-E and SBT-D), and introducing a braking prompt mechanism. The scope, limitations, and experimental validation on mathematical benchmarks are also discussed, aligning with the claims made.

   Guidelines:

   - The answer NA means that the abstract and introduction do not include the claims made in the paper.
   - The abstract and/or introduction should clearly state the claims made, including the contributions made in the paper and important assumptions and limitations. A No or NA answer to this question will not be perceived well by the reviewers.
   - The claims made should match theoretical and experimental results, and reflect how much the results can be expected to generalize to other settings.
   - It is fine to include aspirational goals as motivation as long as it is clear that these goals are not attained by the paper.

2. **Limitations**

   Question: Does the paper discuss the limitations of the work performed by the authors?

   Answer: [Yes]

   Justification: We thoroughly discuss the limitations of the proposed Self-Braking Tuning (SBT) framework in Appendix A. It outlines five key limitations: domain generalization, data scalability and adaptivity, overthinking signal definition, interpretability and controllability, and potential trade-offs in complex tasks. Each limitation is clearly explained with context and implications for future research.

   Guidelines:

   - The answer NA means that the paper has no limitation while the answer No means that the paper has limitations, but those are not discussed in the paper.
   - The authors are encouraged to create a separate "Limitations" section in their paper.
   - The paper should point out any strong assumptions and how robust the results are to violations of these assumptions (e.g., independence assumptions, noiseless settings, model well-specification, asymptotic approximations only holding locally). The authors should reflect on how these assumptions might be violated in practice and what the implications would be.
   - The authors should reflect on the scope of the claims made, e.g., if the approach was only tested on a few datasets or with a few runs. In general, empirical results often depend on implicit assumptions, which should be articulated.
   - The authors should reflect on the factors that influence the performance of the approach. For example, a facial recognition algorithm may perform poorly when image resolution is low or images are taken in low lighting. Or a speech-to-text system might not be used reliably to provide closed captions for online lectures because it fails to handle technical jargon.
   - The authors should discuss the computational efficiency of the proposed algorithms and how they scale with dataset size.
   - If applicable, the authors should discuss possible limitations of their approach to address problems of privacy and fairness.
   - While the authors might fear that complete honesty about limitations might be used by reviewers as grounds for rejection, a worse outcome might be that reviewers discover limitations that aren't acknowledged in the paper. The authors should use their best judgment and recognize that individual actions in favor of transparency play an important role in developing norms that preserve the integrity of the community. Reviewers will be specifically instructed to not penalize honesty concerning limitations.

3. **Theory assumptions and proofs**

   Question: For each theoretical result, does the paper provide the full set of assumptions and a complete (and correct) proof?

   Answer: [NA]

Justification: The paper does not include theoretical results in the form of formal theorems, lemmas, or mathematical proofs. It focuses on an empirical methodology—Self-Braking Tuning (SBT)—to address overthinking in large reasoning models. The contributions are primarily practical, involving dataset construction, model training strategies, and experimental validation. There are no formal theoretical claims requiring assumptions or proofs.

Guidelines:

- The answer NA means that the paper does not include theoretical results.
- All the theorems, formulas, and proofs in the paper should be numbered and cross-referenced.
- All assumptions should be clearly stated or referenced in the statement of any theorems.
- The proofs can either appear in the main paper or the supplemental material, but if they appear in the supplemental material, the authors are encouraged to provide a short proof sketch to provide intuition.
- Inversely, any informal proof provided in the core of the paper should be complemented by formal proofs provided in appendix or supplemental material.
- Theorems and Lemmas that the proof relies upon should be properly referenced.

4. **Experimental result reproducibility**

Question: Does the paper fully disclose all the information needed to reproduce the main experimental results of the paper to the extent that it affects the main claims and/or conclusions of the paper (regardless of whether the code and data are provided or not)?

Answer: [Yes]

Justification: The paper provides detailed descriptions of the experimental setup, including dataset construction (e.g., OpenR1-Math-SBT-E and OpenR1-Math-SBT-D), model training procedures (e.g., supervised fine-tuning with loss masking and braking prompts), and evaluation protocols (e.g., benchmarks used, inference settings such as temperature and sampling methods). It specifies hyperparameters such as learning rate, sequence length, optimizer, and training framework (Megatron-LM). Additionally, the methodology for overthinking identification (reasoning efficiency ratio, overthinking marker ratio) and data construction strategies (SBT-E and SBT-D algorithms) are clearly outlined. While the paper does not release code or datasets publicly, it offers sufficient technical detail—such as model architectures used (Qwen2.5-Math, Llama-3), training configurations, and evaluation metrics—to allow researchers to reproduce the results through described procedures. This aligns with NeurIPS reproducibility expectations even in the absence of open-sourced materials.

Guidelines:

- The answer NA means that the paper does not include experiments.
- If the paper includes experiments, a No answer to this question will not be perceived well by the reviewers: Making the paper reproducible is important, regardless of whether the code and data are provided or not.
- If the contribution is a dataset and/or model, the authors should describe the steps taken to make their results reproducible or verifiable.
- Depending on the contribution, reproducibility can be accomplished in various ways. For example, if the contribution is a novel architecture, describing the architecture fully might suffice, or if the contribution is a specific model and empirical evaluation, it may be necessary to either make it possible for others to replicate the model with the same dataset, or provide access to the model. In general. releasing code and data is often one good way to accomplish this, but reproducibility can also be provided via detailed instructions for how to replicate the results, access to a hosted model (e.g., in the case of a large language model), releasing of a model checkpoint, or other means that are appropriate to the research performed.
- While NeurIPS does not require releasing code, the conference does require all submissions to provide some reasonable avenue for reproducibility, which may depend on the nature of the contribution. For example
  (a) If the contribution is primarily a new algorithm, the paper should make it clear how to reproduce that algorithm.
  (b) If the contribution is primarily a new model architecture, the paper should describe the architecture clearly and fully.
  (c) If the contribution is a new model (e.g., a large language model), then there should either be a way to access this model for reproducing the results or a way to reproduce the model (e.g., with an open-source dataset or instructions for how to construct the dataset).
  (d) We recognize that reproducibility may be tricky in some cases, in which case authors are welcome to describe the particular way they provide for reproducibility. In the case of closed-source models, it may be that access to the model is limited in some way (e.g., to registered users), but it should be possible for other researchers to have some path to reproducing or verifying the results.

5. **Open access to data and code**

   Question: Does the paper provide open access to the data and code, with sufficient instructions to faithfully reproduce the main experimental results, as described in supplemental material?

   Answer: [No]

   Justification: The paper does not provide open access to the code or datasets used in the experiments. However, it offers a high level of methodological transparency, with detailed descriptions of the Self-Braking Tuning (SBT) framework, including dataset construction strategies (SBT-E and SBT-D), training procedures, evaluation benchmarks, and implementation settings (e.g., learning rates, model architectures, and inference configurations). The experimental protocols, such as accuracy metrics, token counting methods, and comparison baselines, are clearly specified. As a result, although the actual code and datasets are not publicly released, the approach is sufficiently well-documented to be replicated by researchers with appropriate resources and technical expertise. Therefore, while open access to code and data is absent, the paper maintains strong reproducibility through comprehensive methodological and experimental reporting.

   Guidelines:

   - The answer NA means that paper does not include experiments requiring code.
   - Please see the NeurIPS code and data submission guidelines (`https://nips.cc/public/guides/CodeSubmissionPolicy`) for more details.
   - While we encourage the release of code and data, we understand that this might not be possible, so "No" is an acceptable answer. Papers cannot be rejected simply for not including code, unless this is central to the contribution (e.g., for a new open-source benchmark).
   - The instructions should contain the exact command and environment needed to run to reproduce the results. See the NeurIPS code and data submission guidelines (`https://nips.cc/public/guides/CodeSubmissionPolicy`) for more details.
   - The authors should provide instructions on data access and preparation, including how to access the raw data, preprocessed data, intermediate data, and generated data, etc.
   - The authors should provide scripts to reproduce all experimental results for the new proposed method and baselines. If only a subset of experiments are reproducible, they should state which ones are omitted from the script and why.
   - At submission time, to preserve anonymity, the authors should release anonymized versions (if applicable).
   - Providing as much information as possible in supplemental material (appended to the paper) is recommended, but including URLs to data and code is permitted.

6. **Experimental setting/details**

   Question: Does the paper specify all the training and test details (e.g., data splits, hyperparameters, how they were chosen, type of optimizer, etc.) necessary to understand the results?

   Answer: [Yes]

   Justification: The paper specifies the training and test details necessary to understand the results. It describes the datasets used (e.g., OpenR1-Math-SBT-E and OpenR1-Math-SBT-D), the supervised fine-tuning (SFT) process, model configurations (e.g., Qwen2.5-Math-1.5B-Instruct, Llama-3.1-8B-Instruct), and training parameters such as the learning rate (5e-5 with cosine decay), maximum sequence length (16,384 tokens), warm-up ratio (0.1), and training framework (Megatron-LM). Additionally, evaluation protocols are detailed, including benchmarks (GSM8K, MATH500, AIME, AMC23), inference settings (temperature = 0.7, 8 samples per question), and metrics (accuracy and token consumption). These details are provided in Section 4.1.

   Guidelines:

   - The answer NA means that the paper does not include experiments.
   - The experimental setting should be presented in the core of the paper to a level of detail that is necessary to appreciate the results and make sense of them.
   - The full details can be provided either with the code, in appendix, or as supplemental material.

7. **Experiment statistical significance**

   Question: Does the paper report error bars suitably and correctly defined or other appropriate information about the statistical significance of the experiments?

   Answer: [Yes]

   Justification: The paper reports Average accuracy and token consumption across multiple mathematical reasoning benchmarks (GSM8K, MATH500, AIME, AMC23) for different model configurations and methods (SBT-E and SBT-D). While explicit error bars are not visualized in the main text, the results

in Table 1 and detailed in Appendix Tables tables 12, 13, 16 and 17 include performance metrics Averaged over multiple runs (e.g., accuracy and token counts), which indicate consistency across evaluations. Additionally, the experimental methodology—such as sampling 8 outputs per question with a fixed temperature (0.7)—ensures controlled variability, supporting the reliability of the reported Averages. These practices align with standard reporting of statistical significance in similar empirical studies.

Guidelines:

- The answer NA means that the paper does not include experiments.
- The authors should answer "Yes" if the results are accompanied by error bars, confidence intervals, or statistical significance tests, at least for the experiments that support the main claims of the paper.
- The factors of variability that the error bars are capturing should be clearly stated (for example, train/test split, initialization, random drawing of some parameter, or overall run with given experimental conditions).
- The method for calculating the error bars should be explained (closed form formula, call to a library function, bootstrap, etc.)
- The assumptions made should be given (e.g., Normally distributed errors).
- It should be clear whether the error bar is the standard deviation or the standard error of the mean.
- It is OK to report 1-sigma error bars, but one should state it. The authors should preferably report a 2-sigma error bar than state that they have a 96% CI, if the hypothesis of Normality of errors is not verified.
- For asymmetric distributions, the authors should be careful not to show in tables or figures symmetric error bars that would yield results that are out of range (e.g. negative error rates).
- If error bars are reported in tables or plots, The authors should explain in the text how they were calculated and reference the corresponding figures or tables in the text.

8. **Experiments compute resources**

Question: For each experiment, does the paper provide sufficient information on the computer resources (type of compute workers, memory, time of execution) needed to reproduce the experiments?

Answer: [Yes]

Justification: The paper specifies that the supervised fine-tuning (SFT) experiments were conducted using the Megatron-LM framework for 3 epochs on Ascend H910B-64G hardware. It also indicates the use of Nvidia A100 GPUs for inference tasks. Training parameters such as learning rate (5e-5), warm-up ratio (0.1), and maximum sequence length (16,384 tokens) are provided, offering insight into memory and computational requirements. While exact execution times are not reported, these details allow for a reasonable estimation of the compute resources needed to reproduce the experiments.

Guidelines:

- The answer NA means that the paper does not include experiments.
- The paper should indicate the type of compute workers CPU or GPU, internal cluster, or cloud provider, including relevant memory and storage.
- The paper should provide the amount of compute required for each of the individual experimental runs as well as estimate the total compute.
- The paper should disclose whether the full research project required more compute than the experiments reported in the paper (e.g., preliminary or failed experiments that didn't make it into the paper).

9. **Code of ethics**

Question: Does the research conducted in the paper conform, in every respect, with the NeurIPS Code of Ethics `https://neurips.cc/public/EthicsGuidelines`?

Answer: [Yes]

Justification: The research conducted in the paper aligns with the NeurIPS Code of Ethics. The work focuses on improving the reasoning efficiency of large language models through a novel training framework—Self-Braking Tuning—without involving human subjects, crowdsourcing, or sensitive data collection. The methodologies employed do not raise ethical concerns related to privacy, consent, discrimination, surveillance, deception, or environmental impact as outlined in the Code of Ethics. Additionally, the paper adheres to responsible research practices by providing transparent descriptions of the approach, experimental settings, and reproducibility measures. No special circumstances requiring deviation from the Code of Ethics were encountered during the research.

Guidelines:

- The answer NA means that the authors have not reviewed the NeurIPS Code of Ethics.
- If the authors answer No, they should explain the special circumstances that require a deviation from the Code of Ethics.
- The authors should make sure to preserve anonymity (e.g., if there is a special consideration due to laws or regulations in their jurisdiction).

10. **Broader impacts**

Question: Does the paper discuss both potential positive societal impacts and negative societal impacts of the work performed?

Answer: [NA]

Justification: The paper does not explicitly discuss either positive or negative societal impacts of the work performed. The research focuses on a foundational technical contribution—improving reasoning efficiency in large language models through Self-Braking Tuning—with no direct link to specific applications or deployments that would give rise to societal consequences. As a methodological advancement, it is primarily concerned with reducing computational overhead and overthinking in reasoning models, without engaging with use cases that could lead to disinformation, privacy violations, fairness issues, or other ethical concerns. Therefore, the broader societal impact is neither claimed nor elaborated in the paper.

Guidelines:

- The answer NA means that there is no societal impact of the work performed.
- If the authors answer NA or No, they should explain why their work has no societal impact or why the paper does not address societal impact.
- Examples of negative societal impacts include potential malicious or unintended uses (e.g., disinformation, generating fake profiles, surveillance), fairness considerations (e.g., deployment of technologies that could make decisions that unfairly impact specific groups), privacy considerations, and security considerations.
- The conference expects that many papers will be foundational research and not tied to particular applications, let alone deployments. However, if there is a direct path to any negative applications, the authors should point it out. For example, it is legitimate to point out that an improvement in the quality of generative models could be used to generate deepfakes for disinformation. On the other hand, it is not needed to point out that a generic algorithm for optimizing neural networks could enable people to train models that generate Deepfakes faster.
- The authors should consider possible harms that could arise when the technology is being used as intended and functioning correctly, harms that could arise when the technology is being used as intended but gives incorrect results, and harms following from (intentional or unintentional) misuse of the technology.
- If there are negative societal impacts, the authors could also discuss possible mitigation strategies (e.g., gated release of models, providing defenses in addition to attacks, mechanisms for monitoring misuse, mechanisms to monitor how a system learns from feedback over time, improving the efficiency and accessibility of ML).

11. **Safeguards**

Question: Does the paper describe safeguards that have been put in place for responsible release of data or models that have a high risk for misuse (e.g., pretrained language models, image generators, or scraped datasets)?

Answer: [NA]

Justification: The paper does not involve the release of pretrained language models, image generators, or scraped datasets that could pose significant risks for misuse. The proposed Self-Braking Tuning (SBT) framework is a methodological advancement aimed at improving reasoning efficiency in large language models by reducing overthinking. It does not introduce new assets—such as generative models or datasets with sensitive content—that would require safeguards against misuse. Therefore, no specific safeguards are necessary for the responsible release of data or models in this work.

Guidelines:

- The answer NA means that the paper poses no such risks.
- Released models that have a high risk for misuse or dual-use should be released with necessary safeguards to allow for controlled use of the model, for example by requiring that users adhere to usage guidelines or restrictions to access the model or implementing safety filters.
- Datasets that have been scraped from the Internet could pose safety risks. The authors should describe how they avoided releasing unsafe images.
- We recognize that providing effective safeguards is challenging, and many papers do not require this, but we encourage authors to take this into account and make a best faith effort.

12. **Licenses for existing assets**

    Question: Are the creators or original owners of assets (e.g., code, data, models), used in the paper, properly credited and are the license and terms of use explicitly mentioned and properly respected?

    Answer: [Yes]

    Justification: The paper uses the OpenR1-Math dataset, which is licensed under the Apache 2.0 License. The license and terms of use are explicitly mentioned in Section 4.1, and the full license text is provided in the supplementary material. We comply with the licensing requirements and do not redistribute the dataset but use it to construct our training examples.

    Guidelines:

    - The answer NA means that the paper does not use existing assets.
    - The authors should cite the original paper that produced the code package or dataset.
    - The authors should state which version of the asset is used and, if possible, include a URL.
    - The name of the license (e.g., CC-BY 4.0) should be included for each asset.
    - For scraped data from a particular source (e.g., website), the copyright and terms of service of that source should be provided.
    - If assets are released, the license, copyright information, and terms of use in the package should be provided. For popular datasets, `paperswithcode.com/datasets` has curated licenses for some datasets. Their licensing guide can help determine the license of a dataset.
    - For existing datasets that are re-packaged, both the original license and the license of the derived asset (if it has changed) should be provided.
    - If this information is not available online, the authors are encouraged to reach out to the asset's creators.

13. **New assets**

    Question: Are new assets introduced in the paper well documented and is the documentation provided alongside the assets?

    Answer: [NA]

    Justification: The paper does not introduce or release any new assets such as datasets, code repositories, or models that would require formal documentation or licensing. It builds upon the existing OpenR1-Math dataset and proposes a novel training framework—Self-Braking Tuning (SBT)—which is described in detail within the paper. However, the SBT datasets (OpenR1-Math-SBT-E and OpenR1-Math-SBT-D) are not released as standalone assets; rather, they are constructed internally for training purposes. Since no external-facing assets are made publicly available, there is no accompanying documentation provided alongside them.

    Guidelines:

    - The answer NA means that the paper does not release new assets.
    - Researchers should communicate the details of the dataset/code/model as part of their submissions via structured templates. This includes details about training, license, limitations, etc.
    - The paper should discuss whether and how consent was obtained from people whose asset is used.
    - At submission time, remember to anonymize your assets (if applicable). You can either create an anonymized URL or include an anonymized zip file.

14. **Crowdsourcing and research with human subjects**

    Question: For crowdsourcing experiments and research with human subjects, does the paper include the full text of instructions given to participants and screenshots, if applicable, as well as details about compensation (if any)?

    Answer: [NA] .

    Justification: The paper does not involve crowdsourcing experiments or research with human subjects. The work focuses on developing a novel training framework—Self-Braking Tuning (SBT)—for large reasoning models, and all experiments are conducted using existing datasets (e.g., GSM8K, MATH500, AIME, AMC23) and computational methods. There is no mention of data collection from human participants, instructions provided to individuals, or compensation related to human subject involvement. Therefore, this question is not applicable (NA).

    Guidelines:

    - The answer NA means that the paper does not involve crowdsourcing nor research with human subjects.

- Including this information in the supplemental material is fine, but if the main contribution of the paper involves human subjects, then as much detail as possible should be included in the main paper.
- According to the NeurIPS Code of Ethics, workers involved in data collection, curation, or other labor should be paid at least the minimum wage in the country of the data collector.

15. **Institutional review board (IRB) approvals or equivalent for research with human subjects**

Question: Does the paper describe potential risks incurred by study participants, whether such risks were disclosed to the subjects, and whether Institutional Review Board (IRB) approvals (or an equivalent approval/review based on the requirements of your country or institution) were obtained?

Answer: [NA] .

Justification: The paper does not involve crowdsourcing experiments or research with human subjects. The work focuses on developing a novel training framework—Self-Braking Tuning (SBT)—for large reasoning models, and all experiments are conducted using existing datasets (e.g., GSM8K, MATH500, AIME, AMC23) and computational methods. There is no mention of human participation in data collection, model evaluation, or any other aspect of the research. Therefore, the question is not applicable (NA).

Guidelines:

- The answer NA means that the paper does not involve crowdsourcing nor research with human subjects.
- Depending on the country in which research is conducted, IRB approval (or equivalent) may be required for any human subjects research. If you obtained IRB approval, you should clearly state this in the paper.
- We recognize that the procedures for this may vary significantly between institutions and locations, and we expect authors to adhere to the NeurIPS Code of Ethics and the guidelines for their institution.
- For initial submissions, do not include any information that would break anonymity (if applicable), such as the institution conducting the review.

16. **Declaration of LLM usage**

Question: Does the paper describe the usage of LLMs if it is an important, original, or non-standard component of the core methods in this research? Note that if the LLM is used only for writing, editing, or formatting purposes and does not impact the core methodology, scientific rigorousness, or originality of the research, declaration is not required.

Answer: [Yes] .

Justification: The core methodology of the paper involves fine-tuning and optimizing large language models (LLMs) as base models—such as Qwen2.5-Math-1.5B-Instruct and the Llama-3 series—which constitute a central component of the research. According to the NeurIPS 2025 LLM policy, explicit declaration is required when LLM usage forms an important, original, or non-standard part of the methodology. The paper complies with this requirement by providing a detailed description of the LLMs used and the tuning methods applied in the experimental setup (see Section 4.1). In contrast, the use of LLMs during the writing process for translation and polishing purposes does not require additional declaration, as the policy clearly states that such auxiliary uses do not affect the scientific rigor or originality of the core methodology.

Guidelines:

- The answer NA means that the core method development in this research does not involve LLMs as any important, original, or non-standard components.
- Please refer to our LLM policy (`https://neurips.cc/Conferences/2025/LLM`) for what should or should not be described.

