# OpenReview forum: "Let LRMs Break Free from Overthinking via Self-Braking Tuning"
_NeurIPS.cc/2025/Conference — NeurIPS 2025 poster_

### Official Review · Reviewer_FmRF · 2025-06-09

**Clarity:** 2
**Significance:** 2
**Originality:** 2
**Rating:** 3
**Confidence:** 5

**Summary:**

This paper aims to study and address the issue of overthinking in current large reasoning models (LRMs). The authors first introduce two metrics — Reasoning Efficiency Ratio and Overthinking Marker Ratio — to quantitatively measure the extent of overthinking in models. Based on these metrics, they propose an adaptive data processing pipeline to construct Self-Braking Tuning datasets, which are designed to teach the model to stop reasoning at the appropriate time. The goal is to improve the model’s inference efficiency after training on the constructed datasets, without causing overthinking. Experimental results show that models trained on the Self-Braking Tuning datasets cost fewer tokens than the baseline model trained on the original o1-like datasets does.

**Questions:**

Please refer to the weakness part above.

**Ethical Concerns:**

["NO or VERY MINOR ethics concerns only"]

**Final Justification:**

I am still concerned about the methodology. I am concerned that the proposed method cannot truly achieve adaptive thinking because of the hard constraints on the constructed training data. I am staying negative.

**Limitations:**

yes

**Paper Formatting Concerns:**

1. Table captions should be listed above the table.

2. Headings should be lower-cased (except for first character).

I suggest the authors to carefully read the format instructions to avoid these presentations errors.

**Quality:**

2

**Strengths And Weaknesses:**

**Strengths**:

(s1) The paper can be easily followed, and Figure 2 well describes the entire pipeline.

(s2) Addressing overthinking issue is a widely studied problem.

**Weaknesses**:

(w1) First, regarding the two metrics proposed to measure the overthinking issue: Reasoning Efficiency Ratio has already been introduced in prior work [1], and Overthinking Marker Ratio is not a very reliable metric (as acknowledge by the authors). Therefore, this part of the contribution is weak. Moreover, the authors do not clearly clarify how the reasoning steps are segmented for calculating the Reasoning Efficiency Ratio.

(w2) Then, the Overthinking Score is also quite heuristic. The choice of $\beta$ = 0.1 also lacks sufficient and convincing justification, and there is no accompanying ablation study to support this design decision.

(w3) Regarding the data construction procedure, it seems that Self-Braking Tuning Exact appears to rigidly retain exactly two full rounds of reasoning in each reasoning path, which seems overly simplistic. Meanwhile, Self-Braking Tuning Dynamic also relies on manually predefined thresholds to determine truncation points. They do not fully align with the authors’ original motivation: enabling the model itself  to autonomously decide when to stop reasoning.

(w4) Regarding the empirical results, though the proposed method can effectively reduce reasoning tokens, the test-time scaling performance is worse than the baseline in most settings in Table 1. Maintaining or improving model performance should take precedence over merely reducing token usage.

(w5) I do not clearly understand the purpose of adding Natural Language Guidance at the reasoning stop points. Moreover, Table 5 shows that adding Natural Language Guidance and using Special Tokens yield similar performance.

(w6) The ablation or convincing explanation on Masked Redundant Thinking  practice is missing, which makes it difficult to assess its actual contribution and necessity.

[1] Chen, Xingyu, et al. "Do not think that much for 2+ 3=? on the overthinking of o1-like llms." ICML 2025

---

> ### Author Rebuttal · Authors · 2025-07-31
>
> Dear Reviewer FmRF,
>
> We thank you for your thorough review and constructive feedback. We have tried to address each of your concerns point by point in the rebuttal.
>
> **Q1: About the Design of the Overthink Score**
>
> **A1**：
>
> **1. About the relation to prior ξ$_O$ metric [1]**:
> We acknowledge that [1] is a valuable work, which we also cite.  However, our formulation introduces two key enhancements better suited for our Self-Braking Tuning framework:
>
> - **Purpose**: Unlike [1], we leverage RER not just for post-hoc analysis but as a core quantitative signal for data construction(Section 2.3), enabling step-wise adaptive truncation based on per-example overthink scores.
>
> - **Granularity**: Their ξ$_O$  operates at the token level, whereas ours works at the **reasoning step level**, respecting logical boundaries.   Our ablations confirm this structural granularity better supports overthinking mitigation (Section 4.3).
>
> **2. About the Reliability of OMR**
>
> **a. OMR as an intentionally sensitive signal:** We do not consider OMR unreliable, but rather intentionally sensitive to capture subtle linguistic cues of overthinking such as "Wait" or "Alternatively."  Prior works [2,3] support using such markers to identify reasoning drift.
>
> **b. Balancing OMR's contribution:** To address potential noise, we assign OMR a low weight in the Overthink Score calculation, ensuring it informs without dominating the objective.
>
> **c. Ablation validation:** Removing OMR entirely shows its clear contribution:
> ||Accuracy|Tokens|
> |-|-|-|
> |w/ OMR|57.83%|1673|
> |w/o OMR|56.25%|2071|
>
> OMR enhances both accuracy (+1.58%) and efficiency (-398 tokens) by capturing linguistic signals that RER alone misses, forming a complementary structural-linguistic detection mechanism.
>
> **3. On reasoning step segmentation:**
> Reasoning steps are segmented by double newlines in model outputs—a widely validated approach in the literature [4,5]. Each segment separated by \n\n represents one reasoning step.
>
> [1] "Do not think that much for 2+ 3=?  on the overthinking of o1-like llms."ICML 2025
>
> [2] "Demystifying long chain-of-thought reasoning in LLMs."  ICLR 2025
>
> [3] "Retro-search: Exploring untaken paths for deeper and efficient reasoning." arXiv:2504.04383
>
> [4] "Can Large Language Models Detect Errors in Long Chain-of-Thought Reasoning."  ACL 2025
>
> [5] "The lessons of developing process reward models in mathematical reasoning."  ACL 2025
>
> **Q2:About the Overthink Score**
>
> **A2:**
>
> **(1) On the Design Motivation for Setting β = 0.1 in the Overthink Score**
>
> We set β = 0.1 to prioritize structural reliability (ηs) while preserving linguistic sensitivity (κt).  The 90:10 weighting ensures stability across different prompting styles while capturing subtle overthinking signals. Our ablation study validates this choice, showing consistent performance across threshold values.
>
> **(2) On the Empirical Justification for β = 0.1 through Ablation Study**
>
> To support this design choice, we conducted an ablation study by varying β ∈ {0.05, 0.1, 0.15, 0.2} for both SBT-E and SBT-D truncation strategies. The results are shown below:
>
> |Method|β|Accuracy|Tokens|
> |-|-|-|-|
> |SBT-E|0.05|56.48%|1762|
> | |0.1|**57.83%**|**1673**|
> | |0.15|56.52%|1874|
> | |0.2|55.86%|1809|
> |SBT-D|0.05|56.24%|**1678**|
> | |0.1|**56.66%**|1682|
> | |0.15|56.21%|1784|
> | |0.2|55.74%|1814|
>
> These results show β = 0.1 yields optimal accuracy-efficiency trade-offs across both methods. Higher β values cause accuracy degradation, confirming our parameter choice.
>
> **Q3: About the SBT data construction procedure**
>
> **A3**：
> We would like to clarify that **SBT-Exact** and **SBT-Dynamic** are not final inference strategies, but **data construction methods** designed to guide the model toward learning when to stop reasoning.
>
> **(1) About the design of SBT-Exact:**
>
> We address your concern through both existing results from Section 4.2 and additional experiments.  The systematic comparison across different solution counts demonstrates:
>
> |Solutions Retained|Accuracy|Tokens|
> |-|-|-|
> | 1 solution|56.95%|1700|
> | 2 solutions|**57.83%**|**1673**|
> | 3 solutions|57.81%|1771|
>
> This analysis reveals that **retaining two complete solutions provides the optimal balance**: sufficient reasoning exposure for robust learning while establishing clear termination signals through answer convergence.
>
> **(2) About the design of SBT-Dynamic**:
> SBT-D uses τ₁=0.2 based on systematic evaluation across multiple values, identifying about 60% of samples as overthinking.  The step-wise evaluation enables **problem-specific adaptive termination**—simple problems stop after 3-4 steps while complex ones continue for 15-20 steps, determined by the model's own reasoning patterns rather than fixed cutoffs.
>
> **(3) About spontaneous overthinking control**:
>
> These augmentations develop **model self-awareness of overthinking**. After SBT, generation becomes **fully autonomous**—models learn to shorten outputs and signal stopping without external control (Fig. 1&2). Analysis of **Qwen2.5-Math-7B-SBT-E** on AIME shows substantial outputs naturally terminate early via epiphany sentences, achieving both higher efficiency and accuracy.
>
> ||Type|% of Cases|Acc|Tok|
> |-|-|-|-|-|
> |AIME24|Early Exit|50.83%|41.80%|5692|
> ||No Early Exit|49.17%|38.98%|11084|
> |AIME25|Early Exit|49.17%|41.53%|6483|
> ||No Early Exit|50.83%|32.79%|12201|
>
> These findings further support that the self-braking behavior is not enforced by hard constraints, but emerges naturally during inference—yielding better performance with less computation.
>
> **Q4:About Performance Under SBT**
>
> **A4:**
> We agree that model performance should remain the priority.    Our results reflect an inherent **accuracy-efficiency trade-off**, consistent with prior efficiency-focused work [1,2].
>
> 1.  **Performance preservation strategies:** We actively mitigate performance degradation through several strategies:
>
> (1) **dual supervision** using SBT-Exact and SBT-Dynamic for both fixed and adaptive stopping patterns,
>
> (2) **explicit braking cues** providing natural language signals for sufficient reasoning depth.
>
> (3) **balanced training** mixing full and truncated trajectories to preserve reasoning depth.
>
> Under these measures, we achieve well-controlled performance impact with an average accuracy drop of 1.43% across all models and maximum drop of 2.86%.  Notably, for general models (LLaMA), SBT-D actually improves both accuracy and efficiency, demonstrating the method's potential in broader settings.
>
> 2.  **Key insight**: Baseline models may achieve correctness through redundancy rather than focused reasoning.    Our approach teaches models to reach correct answers more efficiently, with the slight accuracy trade-off reflecting elimination of this redundant computation rather than loss of reasoning capability.
>
> [1] Ma, Xinyin, et al. "CoT-Valve: Length-Compressible Chain-of-Thought Tuning." ACL 2025
>
> [2] Xia, Heming, et al. "TokenSkip: Token Pruning for Vision-Language Models." ACL 2025.
>
> **Q5: On the Purpose and Effectiveness of Natural Language Guidance (NLG)**
>
> **A5:**
>
> 1.  **Essential Role of Explicit Stopping Signals:**
> Without NLG, our training data would consist merely of variable-length reasoning trajectories with abrupt truncations.       This creates a fundamental learning gap: models would have no explicit supervision signal for recognizing when reasoning becomes sufficient, instead passively waiting for external `</think>` markers.  Our ablation study confirms this necessity:
>
> |Configuration|Accuracy|Tokens|
> |-|-|-|
> |w/ Natural Language|56.66%|1682|
> |w/o Natural Language|56.39%|1801|
>
> **Removing guidance reduces accuracy (-0.27%) and increases token consumption (+7.1%)**, demonstrating that explicit metacognitive cues are essential for self-regulation learning.
>
> 2.  **Superior Efficiency Over Special Tokens:**
> While Table 5 shows comparable overall performance, detailed results in Table 9 reveal NLG's advantages on challenging tasks:
> - **AIME**: 3381 vs. 3647 tokens (-7.3%), 14.17% vs. 12.34% accuracy (+14.8%)
> - **Overall**: 1682 vs. 1797 tokens (-6.4% average)
>
> **Performance differences are most pronounced on complex reasoning**, where natural language's semantic richness provides clearer termination cues.
>
> 3.  **Key Advantages of Natural Language Signals:**
> - **Semantic alignment** with pretraining makes them easier to learn than special tokens
> - **Self-awareness** enables internal reasoning state reflection
> - **Fluent integration** allows natural generation during autonomous inference
>
> **Q6: On the necessity and contribution of Masked Redundant Thinking (MRT)**
> **A6:**
> We address your concern in two parts:
>
> 1.  **Foundational Motivation and Supporting Work:**
> MRT builds on established practices: [1] uses masked incorrect reasoning for step-level self-correction, adopted in [2, 3].  These works demonstrate that masking redundant reasoning improves abstraction, robustness, and self-regulation—core objectives of Self-Braking Tuning.
>
> 2.  **Empirical Evidence and Ablation Study:**
> Section 4.2 shows shorter masked spans improve reasoning efficiency.  An ablation removing MRT yields:
>
> ||Accuracy|Tokens|
> |-|-|-|
> |w/ MRT|57.83%|1673|
> |w/o MRT|58.02%|2306|
>
> While removing MRT marginally increases accuracy (+0.19%), it significantly raises token consumption (+37.84%), confirming **MRT's essential role in efficiency gains without compromising effectiveness**.
>
> [1] "S³cmath: Spontaneous step-level self-correction makes large language models better mathematical reasoners."  AAAI 2025
>
> [2] "Masked Thought: Simply Masking Partial Reasoning Steps Can Improve Mathematical Reasoning Learning of Language Models."  ACL 2024
>
> [3] "Token Cleaning: Fine-Grained Data Selection for LLM Supervised Fine-Tuning."  ICML 2025
>
> **Q7: On Paper Formatting**
>
> **A7:**
> We thank the reviewer for pointing out the formatting issues. We will address both concerns in the revised manuscript to ensure full compliance with the conference's guidelines.

---

> > ### Comment · Reviewer_FmRF · 2025-08-05
> >
> > I thank the authors' detailed response. However, I still have concerns regarding the main contribution of the paper — the proposed Self-Breaking data construction pipeline.
> >
> > First, for SBT-Exact, retain exactly two complete solutions still seems suboptimal. Although the authors conduct experiments on truncating different numbers of solutions and find that retaining two works well on average across benchmarks, this strategy still lacks adaptability and does not reflect problem-specific reasoning needs.
> >
> > Second, regarding SBT-Dynamic, the authors claim that "simple problems stop after 3–4 steps while complex ones continue for 15–20 steps." Is there any quantitative analysis or case study to support this claim? From what I observe in Figures 1 and 2, and the results in Table 1, the token usage of SBT-Dynamic does not differ significantly from that of SBT-Exact — for example, on the AIME dataset.
> >
> > Also, another point is, the performance gaps between different methods in ablations are quite small, which makes it hard to say whether the strategy really improves the performance.

---

> > > ### Author Response · Authors · 2025-08-06
> > > **1/3  On the adaptability of SBT-E**
> > >
> > > Dear Reviewer FmRF,
> > >
> > > Thank you for your thoughtful follow-up. We appreciate the opportunity to clarify these important points.
> > >
> > > **1. On the adaptability of SBT-E**
> > > We appreciate this thoughtful concern and would like to address the adaptability issue directly.   **The key insight: Problem-specific adaptation occurs WITHIN the two-solution framework, not despite it.** While the NUMBER of solutions remains constant, their internal complexity varies dramatically with problem difficulty.
> > >
> > > **(1). Quantitative Evidence of Internal Adaptation:**
> > >
> > > Since OpenR1-Math-97K does not have explicit difficulty annotations, we use Deepseek-R1-0528 to generate 30 trajectories on datasets of varying difficulty levels (GSM8K, MATH500, AIME24&25) and systematically analyze them..
> > >
> > > |Dataset|Avg Foundation Solution Steps(Tokens)| Avg Evolution Solution Steps(Tokens) |
> > > |-|-|-|
> > > |GSM8K|7.60(110.2)|3.17(44.4)|
> > > |MATH500|9.20(283.4)|3.03(73.7)|
> > > |AIME|14.17(1037.0)|6.43(277.7)|
> > >
> > > This substantial variation (7.6→14.17 steps, 110→1037 tokens) within the same two-solution structure demonstrates remarkable adaptability.   Simple problems get concise solutions;   complex problems develop extensive reasoning chains—all while maintaining structural consistency.
> > >
> > > **(2). Why Exactly Two Solutions—Empirical and Theoretical Justification:**
> > >
> > > **Empirically** (Section 4.2), two solutions achieve optimal trade-offs:
> > > - **One solution**: Too abrupt, introduces hard-to-learn stopping signal (56.95% acc, 1700 tokens)
> > > - **Two solutions**: Optimal, enables natural "solve-verify-stop" pattern (57.83% acc, 1673 tokens)
> > > - **Three+ solutions**: Redundant, no accuracy gains (57.81% acc, 1771 tokens)
> > >
> > > **Theoretically**, two solutions mirror natural problem-solving:
> > > - First solution: Exploration and initial solving
> > > - Second solution: Verification and confidence building
> > > - Beyond two: Diminishing returns (overthinking) begin
> > >
> > > Figure 3(b) reveals that models generate multiple solutions regardless of difficulty—Evolution Solutions appear across GSM8K (41%), MATH500 (47%), and AIME (29%).   This universal pattern confirms overthinking is inherent to LRMs, not problem-specific.
> > >
> > > **(3). The Elegance of Structure-Content Separation:**
> > >
> > > Variable solution counts would require difficulty estimation (adding complexity) and create unstable training signals.   Instead, SBT-E achieves **structural simplicity with content complexity**—the framework provides consistency while content provides adaptation.
> > >
> > > **In summary:** The two-solution structure is a design strength, not a limitation.   It serves as a **practical, robust default** that provides consistent learning signals while allowing natural adaptation through variable solution complexity.   This explains why SBT-E, despite its apparent simplicity, achieves competitive performance with the more complex SBT-D approach.

---

> > > ### Author Response · Authors · 2025-08-06
> > > **2/3 Quantitative Evidence for SBT-Dynamic's Adaptive Behavior**
> > >
> > > (Due to space limitations, I will continue responding to your question here.
> > > )
> > >
> > > **2.Quantitative Evidence for SBT-Dynamic's Adaptive Behavior:**
> > >
> > > We appreciate your request for quantitative support.     We need to clarify that our claim "simple problems stop after 3–4 steps while complex ones continue for 15–20 steps" refers to the adaptive phenomenon in SBT-D's data construction, though it is also reflected in our SBT model's inference process.
> > >
> > > **(1). Adaptability in SBT-D Data Construction:**
> > >
> > > We analyzed 90 trajectories from DeepSeek-R1-0528 and applied SBT-D to rapidly construct the corresponding data:
> > >
> > > |Dataset|Original Steps|SBT-D Steps|Avg. Reserved Steps|Reserved (<5) | Reserved (5–15) | Reserved (>15)|
> > > |-|-|-|-|-|-|-|
> > > | GSM8K   | 13.7  | 8.2| 0.6 |93.3% (28)| 6.7% (2)| 0% (0)|
> > > | MATH500 | 19.2| 11.4| 2.2 |70.0% (21)| 26.7% (8)| 3.3% (1)|
> > > | AIME| 31.4| 24.8| 10.6 |46.7% (14)| 20.0% (6)| 33.3% (10)|
> > >
> > >
> > > This demonstrates clear problem-specific adaptation:
> > > - **Simple problems (GSM8K)**: 93.3% terminate within 5 steps after SBT-D processing
> > > - **Complex problems (AIME)**: Only 46.7% terminate early, with 33.3% requiring >15 steps
> > > - **Adaptive reduction**: More aggressive truncation on simpler tasks (0.6, 40%) vs. careful preservation on complex ones (10.6, 21%)
> > >
> > > **(2). Adaptability in SBT Model Inference:**
> > >
> > > We present statistics from Qwen2.5-Math-7B-Instruct-SBT-E's evaluation results (the dramatic step differences likely reflect DeepSeek-R1-0528's optimization, but still demonstrate our framework's capability) :
> > >
> > > | Dataset | AIME25 | AIME24 | AMC23  | MATH500 | GSM8K |
> > > |---------|--------|--------|--------|---------|-------|
> > > | Avg. Steps | 202.23 | 174.97 | 106.40 | 51.32   | 27.78 |
> > >
> > > The **7.3x difference** between GSM8K (27.78 steps) and AIME25 (202.23 steps) clearly shows that our SBT-trained models naturally adapt reasoning length to problem complexity during inference.     This emergent behavior validates that our training framework successfully teaches models to self-regulate based on problem difficulty.
> > >
> > > **(3). On Similar Token Usage Between SBT-D and SBT-E:**
> > >
> > > You are absolutely correct—SBT-D and SBT-E show similar token usage on several benchmarks. **This is by design, not a limitation.**
> > >
> > > The two variants are complementary, not competitive:
> > > - **SBT-E**: Provides structural consistency through uniform two-solution patterns → better training stability and generalization
> > > - **SBT-D**: Offers fine-grained, problem-specific adaptation → captures nuanced reasoning patterns
> > >
> > > The similar token counts (e.g., AIME: 3101 vs 3381) actually **validate our approach**—both methods converge on the same efficiency frontier through different mechanisms.     This suggests we've correctly identified where overthinking begins, whether approached through structural consistency (SBT-E) or adaptive truncation (SBT-D).
> > >
> > > The dual-approach design gives practitioners flexibility: choose SBT-E for simplicity and robustness, or SBT-D for task-specific optimization.     Their similar performance confirms both are effective paths to the same goal—efficient reasoning without overthinking.

---

> > > ### Author Response · Authors · 2025-08-06
> > > **3/3 On Performance Gaps in Ablations**
> > >
> > > (Due to space limitations, I will continue responding to your question here.
> > > )
> > >
> > > **3.On Small Performance Gaps**
> > >
> > > We appreciate the reviewer's attention to the ablation results.  While the performance gaps may appear small in isolation, we believe they demonstrate meaningful improvements when considered holistically.
> > >
> > > **(1) Reframing Success: Efficiency-First Framework**
> > > We emphasize that our primary contribution is achieving **substantial token reduction (30-60%) while maintaining comparable accuracy**.   In this efficiency-focused framework, even small accuracy differences are highly meaningful.   For instance, our step-level vs. token-level ablation (Table 4) demonstrates that step-level detection achieves **56.66% average accuracy versus 56.24% (+0.42%)** while using **1682 versus 1753 tokens (-4.1%)** across all benchmarks.   While these differences may seem modest, they represent crucial insights: (1) even small accuracy gains matter when coupled with efficiency improvements, and (2) on challenging tasks like AIME, the gap widens significantly (14.17% vs. 11.34%, a 25% relative improvement).   This shows that **preserving reasoning coherence through step-level truncation simultaneously improves both accuracy AND efficiency**—the dual optimization that represents our fundamental breakthrough.
> > >
> > > **(2) Systematic Exploration for Optimal Trade-offs**
> > > Our ablation studies systematically validate configurations that consistently achieve these efficiency gains across:
> > > * **Multiple threshold values** (0.2, 0.3, 0.4)
> > > * **Alternative implementations** (step-level vs. token-level, natural language vs. special tokens)
> > > * **Diverse model architectures** (Qwen2.5-Math, Llama-3.1, Llama-3.2)
> > >
> > > This comprehensive evaluation identifies stable parameter settings that work across real-world scenarios without extensive fine-tuning.
> > >
> > > **(3) Small Gaps as Framework Stability**
> > > As the reviewer correctly observes, "the performance gaps between different methods in ablations are quite small."   **We view this as a strength, not a limitation**—it demonstrates the remarkable stability and robustness of our method across varying configurations.   These small differences indicate that SBT provides **consistent and reproducible benefits** regardless of specific settings.
> > >
> > > Consider Table 2: even our **worst-performing threshold setting still achieves 41.5% token reduction** (1917 vs. 3278 tokens) with minimal accuracy loss.   This consistency shows that:
> > > - **Efficiency gains are inherently robust**, not dependent on perfect hyperparameter tuning
> > > - **The framework operates reliably** across a broad range of settings without catastrophic failure modes
> > > - **Practical deployment is straightforward**, working well "out of the box" without extensive optimization
> > >
> > > **In summary, the consistent small accuracy variations (<2%) paired with massive efficiency improvements (30-60%) across all configurations validate that we've achieved the ideal outcome: a robust, generalizable solution that reliably eliminates overthinking without requiring careful tuning or sacrificing reasoning quality.**
> > >
> > > We hope these clarifications address your concerns.  The core insight remains: models can learn to self-regulate reasoning length, achieving substantial efficiency gains with minimal performance impact.
> > >
> > > Best regards,
> > >
> > > The authors of Submission8232

---

> > > ### Author Response · Authors · 2025-08-08
> > >
> > > Dear Reviewer FmRF,
> > >
> > > Thank you once again for your time and thoughtful feedback on our submission.
> > >
> > > We have carefully addressed your concerns regarding the adaptability of SBT-Exact, the empirical support for SBT-Dynamic, and the interpretability of our ablation results in our recent responses.
> > >
> > > As the author-reviewer discussion phase is entering its final stage — **with less than 24 hours remaining** — we would be truly grateful if you could take a moment to review our responses and let us know if you have any remaining questions, concerns, or suggestions.  Your insights have been extremely valuable in helping us strengthen the clarity, rigor, and overall quality of our work, and we deeply value your continued engagement.
> > >
> > > If you find our clarifications satisfactory, we would greatly appreciate it if you would consider re-evaluating your assessment of our submission.
> > >
> > > Thank you again for your constructive input throughout this process.
> > >
> > > Best regards,
> > >
> > > The authors of Submission8232

---

> > > > ### Comment · Reviewer_FmRF · 2025-08-08
> > > >
> > > > I thank the authors for their detailed responses. My major concern regarding the adaptivity of reasoning patterns of SBT-E and SBT-D still remains. The thing is, the number of reasoning tokens is not the key — it's the number of reasoning solutions that matters. A single reasoning solution for a more difficult problem will naturally require more tokens or steps than for an easier one. But the more fundamental question for achieving adaptive thinking is whether more difficult problems also require more reasoning solutions.
> > > >
> > > > Based on the additional results and the discussion, I have decided to update my score to a 3, while I am still concerned about the methodology part.

---

> ### Author Response · Authors · 2025-08-07
>
> Dear Reviewer FmRF,
>
> Thank you once again for your time and thoughtful feedback on our submission.
>
> In our recent responses, We have addressed your concerns regarding the adaptability of SBT-Exact, the empirical evidence supporting SBT-Dynamic, and the interpretability of our ablation results.
>
> As the discussion phase is drawing to a close, we would greatly appreciate it if you could take a moment to review our responses and let us know if you have any further questions or suggestions. Your insights have been extremely valuable in helping us improve the clarity and rigor of our work.
>
> Thank you again for your constructive engagement throughout the review process.
>
> Best regards,
>
> The authors of Submission8232

---

> ### Author Response · Authors · 2025-08-08
>
> **Dear Reviewer FmRF,**
>
> Thank you very much for your continued engagement and for raising a fundamental and insightful question:
>
> > *"But the more fundamental question for achieving adaptive thinking is whether more difficult problems also require more reasoning solutions."*
>
> We fully understand your concern, and would like to address it by analyzing the relationship between **the number of reasoning solutions and reasoning capability** from two complementary perspectives — during **training data construction** and during **inference behavior**.
>
> ---
>
> ### 1.        **On the Number of Solutions in Data Construction**
>
> In our initial rebuttal, we compared the effects of retaining 1, 2, and 3 solutions during SBT-E construction.        Here, we further provide their respective performance on the most challenging dataset — **AIME**:
>
> | Solutions Retained | Accuracy   | Tokens   |
> | ------------------ | ---------- | -------- |
> | 1 solution         | 12.71%     | 3132     |
> | 2 solutions        | **13.75%** | **3101** |
> | 3 solutions        | 12.74%     | 3522     |
>
>
> These results clearly demonstrate that **increasing the number of solutions does not improve performance**;  in fact, retaining 3 solutions **degrades both accuracy and efficiency** compared to retaining 2.    This suggests that increasing the number of reasoning solutions beyond two offers diminishing returns in terms of meaningful reasoning content, and often introduces redundancy.     These results clearly indicate that a greater number of reasoning solutions does not translate into enhanced problem-solving ability.
>
>
> ### 2.        **On the Number of Solutions During Inference**
>
>
>
> As shown in Figure 3(b), we analyzed the average number of solutions generated per question across datasets of varying difficulty using DeepSeek-R1-Distill-Qwen-7B:
>
> * **GSM8K (easiest):** 4.54 solutions, over 1,000 tokens
> * **MATH500 (intermediate):** 4.82 solutions, over 2,000 tokens
> * **AIME (most difficult):** 5.64 solutions, approximately 5,000 tokens
>
> Despite the significant difference in problem difficulty, the difference in the number of generated solutions between the easiest and hardest datasets is, on average, only about one additional solution.     However, this gap is reflected far more substantially in the content — with nearly a **fivefold increase in tokens**, largely concentrated within the **Foundation Solution**.
>
> This suggests that more difficult problems are **not solved by proportionally increasing the number of reasoning solutions**, but rather by **increasing the complexity and depth of each individual solution**.
>
> This directly supports our central design choice in SBT-E:
>
> When we fix the number of reasoning solutions at two, what varies is the level of detail and complexity within each individual solution, depending on the difficulty of the problem.   In other words, **the model adapts by making each reasoning path richer and deeper, rather than by generating more separate reasoning paths.**  This behavior aligns with the reasoning patterns that we have observed in LRMs.  This approach helps maintain stable training while providing the model with enough flexibility to determine when to stop reasoning, without producing excessive redundant or unnecessary solutions.
>
> We hope this analysis helps clarify that the adaptivity in SBT-E and SBT-D does not lie in generating more reasoning solutions for harder problems, but rather in allowing each solution to naturally scale in complexity with problem difficulty — a form of adaptive reasoning that we believe directly addresses your core concern.
>
> We sincerely hope this clarifies the core of your concern, and we are grateful for your thoughtful push to refine the framing of our work.
>
> Best regards,
>
> The authors of Submission8232

---

### Official Review · Reviewer_S6DA · 2025-06-29

**Clarity:** 2
**Significance:** 3
**Originality:** 3
**Rating:** 4
**Confidence:** 4

**Summary:**

This paper addresses the “overthinking” problem in current large reasoning models (LRMs) by proposing a novel framework called Self-Braking Tuning (SBT). The authors introduce several metrics to detect redundant reasoning and leverage these to construct training data with adaptive reasoning lengths, incorporating a braking prompt mechanism. Models fine-tuned using this approach demonstrate significant token cost reduction, highlighting the effectiveness of the proposed framework.

**Questions:**

1. Is this a typo? It seems that the model with the lowest token cost shown in Table 1 — Llama-3.1-8B-Instruct — is marked incorrectly.

**Ethical Concerns:**

["NO or VERY MINOR ethics concerns only"]

**Final Justification:**

After reading the authors' response, I have no further questions. The current scores align with the quality of this paper, so I have decided to maintain it.

**Limitations:**

yes

**Quality:**

3

**Strengths And Weaknesses:**

**Strengths**

1.	The motivation is clear and well-grounded. Constructing data with a braking prompt mechanism presents a reasonable strategy for addressing the overthinking issue.
2.	The results are strong, achieving nearly 50% reduction in token usage while maintaining performance.
3.	The authors contribute a valuable dataset that can serve as a foundation for future research in this area.

**Weaknesses**

1.	The paper is somewhat difficult to follow. One major reason is the introduction of numerous new terms—such as Foundation Solution, Evolution Solution, Reasoning Efficiency Dominance, and Linguistic Indicator Robustness. While I understand the authors’ intention to clarify concepts, these terms could be streamlined or explained in a more accessible way to enhance readability.
2.	The evaluation scope is narrow, focusing exclusively on math-related tasks. Overthinking is not unique to the math domain, so it would be beneficial to include broader benchmarks such as MMLU to demonstrate the framework’s general applicability.
3.	The generalizability of the proposed framework is also a concern. It involves multiple steps and relies on several super-hyperparameters (e.g., thresholds), with no clear guidance on selecting optimal values. Furthermore, the improvements reported in Tables 2–5 appear marginal, raising questions about the robustness and scalability of the method.

---

> ### Author Rebuttal · Authors · 2025-07-31
>
> Dear Reviewer S6DA,
>
> We thank you for your thoughtful review and valuable feedback. We appreciate your recognition of our framework’s motivation, the effectiveness of our results, and the contribution of a useful dataset. We have carefully addressed your comments in the rebuttal.
>
> **Q1: About the new terms**
>
> **A1**：
> We appreciate your feedback on the complexity of our terminology. Our intention was to precisely distinguish between different reasoning concepts and analysis dimensions, but we understand this may have impacted readability. In future versions, we will strive to streamline naming, ensure stylistic consistency, and provide clearer, more intuitive explanations upon first use to improve overall accessibility.
>
> **Q2: About the evaluation scope**
>
> **A2:**
> Thank you for the suggestion.  We understand the importance of evaluating beyond a single domain and offer the following clarification and supporting results:
>
> **1. Focus on the math domain:** Our main evaluation centers on math-related tasks because the SBT framework is trained entirely on math data.  Overthinking behaviors are especially prominent and easier to analyze in this setting, given the well-defined reasoning structure and objective correctness of math problems.  This makes math particularly suitable for developing and initially validating our approach.
>
> **2. Generalization beyond math:** We agree that overthinking is not confined to mathematics.  To test the broader applicability of our method, we additionally evaluate SBT on **two out-of-domain benchmarks**, **MMLU-Redux** and **GPQA-Diamond**, which cover a range of factual and commonsense reasoning challenges.  The results below show that our methods (SBT-E and SBT-D) achieve consistent **token savings** while maintaining competitive accuracy across diverse model backbones—**without any out-of-domain fine-tuning**:
>
> | Base Model| Method   | MMLU-Redux (acc / tokens) | GPQA-Diamond (acc / tokens) | Average (acc / tokens) |
> | - | -| - | -| ---------------------- |
> | Qwen2.5-Math-1.5B-Instruct | baseline | 45.84% / 2061             | 24.75% / 5485               | 35.30% / 3773          |
> |  | sbt-e    | 43.12% / 1403             | 25.06% / 3194               | 34.09% / 2299          |
> |  | sbt-d    | 43.28% / 1566             | 26.20% / 3002               | **34.74%** / **2284**          |
> | Qwen2.5-Math-7B-Instruct   | baseline | 67.04% / 3229             | 41.15% / 8892               | 54.10% / 6061          |
> |  | sbt-e    | 65.84% / 1927             | 40.40% / 6205               | 53.12% / **4066**          |
> |   | sbt-d    | 66.39% / 1998             | 41.29% / 6706               | **53.84%** / 4352          |
> | LLaMA3.2-1B-Instruct       | baseline | 35.62% / 1933             | 17.99% / 9321               | 26.81% / 5627          |
> |  | sbt-e    | 32.24% / 770              | 24.24% / 3516               | 28.24% / 2143          |
> |                            | sbt-d    | 33.12% / 725              | 23.48% / 3157               | **28.30%** / **1941**          |
> | LLaMA3.1-8B-Instruct       | baseline | 80.53% / 2481             | 37.31% / 8918               | 58.92% / 5699          |
> |                            | sbt-e    | 77.46% / 1646             | 36.30% / 5346               | 56.88% / **3496**          |
> |                            | sbt-d    | 77.74% / 1668             | 36.91% / 6717               | **57.33%** / 4192          |
>
> These results confirm that our proposed overthinking-aware strategies transfer well to **non-mathematical reasoning tasks** and generalize across model scales and architectures.  **Specifically, we observe consistent token reductions ranging from 26% to 65% across all model-task combinations, with accuracy drops typically limited to 1-3%**.  Notably, **smaller models (LLaMA3.2-1B) show the largest efficiency gains (65% token reduction) and sometimes even improved accuracy**, while larger models achieve more moderate but still substantial savings (26-40%).  **The fact that math-trained models maintain competitive performance on diverse reasoning tasks without domain-specific retraining demonstrates the fundamental nature of our self-regulation mechanism**—suggesting that learning to recognize reasoning sufficiency is a transferable meta-cognitive skill applicable across reasoning domains.
>
> **Q3: About Framework Generalizability and Improvements**
>
> **A3:**
>
> **1. On Multiple Hyperparameters and Complexity**
>
> Our Self-Braking Tuning framework actually requires **fewer hyperparameters than existing overthinking mitigation approaches**.         While we use two primary thresholds (τ₁ and τ₂), the secondary threshold is simply set as τ₁ + 5%, reducing this to essentially **one core hyperparameter**.         In contrast:
>
> - **RL-based approaches** require reward function design, learning rates, exploration parameters, and length penalty coefficients [1-3]
> - **External constraint methods** need token budgets, stopping criteria, and confidence thresholds [4,5]
> - **Dynamic inference methods** require multiple confidence thresholds, early-stopping parameters, and uncertainty estimation hyperparameters [6-8]
>
> Our approach represents a significant **simplification** in the hyperparameter space while maintaining effectiveness.
>
> **2. On Clear Hyperparameter Selection Guidance**
>
> We agree that guiding parameter selection is essential;       therefore, we dedicate ourselves to providing **systematic empirical guidance** for hyper-parameter choice:
>
> **(1) For threshold τ₁:** Our analysis in Section 4.1 demonstrates that **τ₁ = 0.2 consistently achieves optimal performance** across all model architectures and task difficulties
>
> **(2) For weighting parameter β:** We supplemented our experiments with additional validations that confirmed the choice of β = 0.1.
>
> | Method | β    | Accuracy | Tokens |
> |--------|------|----------|--------|
> | SBT-E  | 0.05 | 56.48%   | 1762   |
> |        | **0.1** | **57.83%** | **1673** |
> |        | 0.15 | 56.52%   | 1874   |
> |        | 0.2  | 55.86%   | 1809   |
> | SBT-D  | 0.05 | 56.24%   | **1678** |
> |        | **0.1** | **56.66%** | 1682   |
> |        | 0.15 | 56.21%   | 1784   |
> |        | 0.2  | 55.74%   | 1814   |
>
> This systematic analysis shows **β = 0.1 consistently achieves the best accuracy-efficiency trade-off** for both SBT variants, providing clear selection guidance rather than arbitrary parameter choices.
>
> **3. On Robustness Demonstrated by Consistent Improvements in Table 2-5**
>
> **(1) Purpose of Comprehensive Ablation Studies:** These ablation studies serve a **critical purpose**: identifying near-optimal configurations that maintain our primary objective of **dramatic efficiency gains (30-60% token reduction)** while preserving accuracy.     Our extensive experiments systematically explore the parameter space across:
>
> - **Multiple threshold values** (0.2, 0.3, 0.4)
> - **Alternative configurations** (step-level vs. token-level, natural language vs. special tokens)
> - **Various model architectures** (Qwen2.5-Math, Llama-3.1, Llama-3.2)
>
> This systematic exploration establishes **optimal operating points** for Self-Braking Tuning across diverse experimental conditions.
>
> **(2) Beyond Peak Performance: Prioritizing Universal Applicability:**
>
> However, our focus extends beyond simply chasing peak numerical performance.     Instead, we aim to develop a **generalizable and stable methodology** that gracefully balances performance and efficiency across diverse conditions.     The consistent patterns observed across all experiments—rather than dramatic variations—indicate that SBT achieves **reliable performance without requiring extensive retuning** for different models or tasks.     This stability is precisely what enables practical deployment and broader applicability.
>
> **(3) Evidence of Framework Robustness:** The reviewer's observation about "marginal improvements" in Tables 2-5 actually demonstrates our framework's **key strength: robustness**.     The **consistency of results** across:
> - **Multiple model architectures**: Qwen2.5-Math, Llama-3.1, Llama-3.2
> - **Varying task difficulties**: GSM8K to AIME
> - **Different experimental configurations**: All ablation combinations
>
> This consistency confirms that our **core mechanism—teaching models autonomous reasoning regulation—is fundamentally sound and transferable**.     This **robustness and reliability** across diverse conditions validates that SBT represents a **mature, deployable solution** rather than a method overfitted to specific experimental settings.
>
> [1] Aggarwal, Pranjal, and Sean Welleck. "L1: Controlling how long a reasoning model thinks with reinforcement learning." arXiv preprint arXiv:2025.
>
> [2] Luo, Haotian, et al. "O1-pruner: Length-harmonizing fine-tuning for o1-like reasoning pruning." arXiv preprint arXiv:2501.12570, 2025.
>
> [3] Shen, Yi, et al. "Dast: Difficulty-adaptive slow-thinking for large reasoning models." arXiv preprint arXiv:2503.04472, 2025.
>
> [4] Han, Tingxu, et al. "Token-budget-aware llm reasoning." arXiv preprint arXiv:2412.18547, 2024.
>
> [5] Xu, Silei, et al. "Chain of draft: Thinking faster by writing less." arXiv preprint arXiv:2502.18600, 2025.
>
> [6] Yang, Chenxu, et al. "Dynamic early exit in reasoning models." arXiv preprint arXiv:2504.15895, 2025.
>
> [7] Zhang, Jintian, et al. "Lightthinker: Thinking step-by-step compression." arXiv preprint arXiv:2502.15589, 2025.
>
> [8] Manvi, Rohin, Anikait Singh, and Stefano Ermon. "Adaptive inference-time compute: Llms can predict if they can do better, even mid-generation." arXiv preprint arXiv:2410.02725, 2024.
>
> > Question: Is this a typo? It seems that the model with the lowest token cost shown in Table 1 — Llama-3.1-8B-Instruct — is marked incorrectly.
>
> Thank you for pointing this out. We must acknowledge that the lowest token cos for LLaMA-3.1-8B-Instruct was incorrectly marked due to an oversight on our part. We appreciate your careful reading and will correct this in the revised version.

---

### Official Review · Reviewer_Bv4y · 2025-06-30

**Clarity:** 3
**Significance:** 2
**Originality:** 3
**Rating:** 4
**Confidence:** 4

**Summary:**

This paper addresses the problem of "overthinking" in Large Reasoning Models (LRMs), where models generate excessively long and redundant chains of thought. The authors propose a novel framework called Self-Braking Tuning (SBT), which teaches LLMs to autonomously regulate their own reasoning process and terminate it at an appropriate point. The core idea is to train the model to develop an "internal braking mechanism" rather than relying on external interventions.

**Questions:**

1. The authors evaluate their framework exclusively on lightweight models that lack inherent long CoT capabilities. It would be informative to assess its performance on models such as DeepSeek Distill, or others that natively support long CoT reasoning.

2. I recommend that the authors include, in the appendix, both illustrative examples and summary statistics demonstrating the differences between the datasets constructed by their strategy and the original datasets. This would give readers a clearer understanding of the framework’s data‑construction impact.

**Ethical Concerns:**

["NO or VERY MINOR ethics concerns only"]

**Final Justification:**

I have read the author's rebuttal and the comments from other reviewers, and after a comprehensive evaluation, I have decided to maintain my original score.

**Limitations:**

yes

**Quality:**

2

**Strengths And Weaknesses:**

**Strengths:**

1. The paper introduces an innovative framework to address the problem of LRM “overthinking.” It comprises three components:

   * A data‑driven method for identifying overthinking instances.
   * An automated, adaptive data‑construction strategy (SBT‑E and SBT‑D).
   * A unique training mechanism that employs redundancy‑content loss masking and natural‑language braking prompts.
     This end‑to‑end process, designed to teach the model self‑regulation, represents a novel contribution to the field.

2. The authors provide a clear analytical breakdown of their approach. In the analysis section, they further elucidate the internal dynamics of overthinking and present fine‑grained mitigation strategies, offering valuable insights and potential directions for future work.

3. The paper is very well written and logically coherent. Terminology is precise, and each component’s motivation within the framework is thoroughly explained.

**Weaknesses:**

1. **Performance on Complex Tasks:** The authors themselves note, in the Limitations section, a potential trade‑off when tackling more complex problems. Experimental results also indicate that, as dataset difficulty increases, performance degrades—suggesting that the framework may, to some extent, impair the model’s self‑correction capabilities, which are a key benefit of longer reasoning chains.

2. **Lack of Module‑Level Ablation Studies:** For example, although the “Natural Language Guidance” module is compared against a special‑token baseline, there is no ablation study omitting this module entirely. This omission makes it difficult to quantify each module’s individual contribution.

3. **Hyperparameter Sensitivity and Usability Concerns:** The framework relies on multiple hyperparameters, raising questions about its scalability and ease of use. The threshold ablation experiments show that optimal values vary across datasets—and in some cases produce trade‑offs between accuracy and token usage—indicating that extensive tuning may be required to achieve peak performance.

---

> ### Author Rebuttal · Authors · 2025-07-31
>
> Dear Reviewer Bv4y,
>
> We thank you for your thorough review and constructive feedback. We sincerely appreciate your positive recognition of the novelty of our framework, the clarity of our analysis, and the overall quality of the writing. We have addressed each of your concerns point by point in the rebuttal.
>
> ---
>
> **W1: About the performance under SBT**
>
> **A1:**
> **1. On the Trade-off Between Accuracy and Efficiency:**
> We would like to clarify that the trade-off between reasoning efficiency and accuracy is not a limitation of our method per se, but rather an inherent challenge in complex multi-step reasoning. Reducing redundant steps can, in some cases, limit the model’s ability to recover answers through repetition or self-correction. This phenomenon is particularly visible in math-specialized models (e.g., Qwen2.5-Math), which rely heavily on deep token-level exploration. Similar trade-offs between accuracy and efficiency have been observed in other studies as well, such as in [1] and [2], further validating this inherent challenge in multi-step reasoning tasks.
>
> **2. About the Accuracy:**
> We recognize that maintaining strong performance is critical, and we adopt multiple strategies to mitigate potential drops. These include diverse supervision (SBT-Exact + SBT-Dynamic), explicit braking prompts, and balanced training between full and early-stopped traces. As a result, the overall performance drop is strictly controlled (≤ 2.86%), with an average gap of only **1.43%** across models—while reducing token usage by up to **60%**.
>
> **3. On General Models:**
> Importantly, SBT is not merely efficient—it also enhances accuracy in some settings. On general-purpose models such as LLaMA,, the SBT-D variant yields consistent gains in both performance and efficiency. This highlights the method’s broader applicability beyond math-heavy tasks, and suggests that SBT can serve as a general-purpose optimization strategy for step-wise reasoning.
>
> [1] "CoT-Valve: Length-Compressible Chain-of-Thought Tuning." ACL 2025.
>
> [2] "TokenSkip: Token Pruning for Vision-Language Models." ACL 2025.
>
> ---
>
> **W2: Addressing Module-Level Ablation Studies**
>
> **A2:**
>
> We'd like to explain our original experimental design and provide the requested module-level evaluation.
>
> **1. Original Experimental Motivation:**
>
> Our initial focus was on comparing different guidance mechanisms (natural language vs. special tokens) rather than evaluating the necessity of guidance signals themselves.    This decision was based on preliminary observations that models without explicit termination cues showed poor self-regulation behavior, making the comparison between guidance types more immediately relevant than the presence vs. absence comparison.
>
> **2. Complete Module Ablation Results:**
>
> We have now conducted the comprehensive ablation study removing natural language guidance entirely:
>
> | Configuration | Accuracy | Tokens | Performance Impact |
> |---------------|----------|--------|--------------------|
> | **w/ Natural Language** | **56.66%** | **1682** | **Baseline** |
> | w/o Natural Language | 56.39% | 1801 | -0.27% acc, +7.1% tokens |
>
> **Key Findings from Complete Ablation:** The removal of natural language guidance results in both **reduced accuracy** (-0.27%) and **increased token consumption** (+7.1%), demonstrating that explicit metacognitive cues provide measurable benefits for both reasoning quality and efficiency.
>
> **3. Individual Module Contributions:**
>
> This complete ablation clarifies each component's role:
> - **Natural Language Guidance**: +0.27% accuracy, -6.6% tokens (vs. no guidance)
> - **Natural Language vs. Special Tokens**: +0.05% accuracy, -6.4% tokens (from Table 9)
>
> The results confirm that explicit termination signals are essential for effective self-regulation, with natural language providing the most efficient implementation of this mechanism.
>
> ---
>
> **W3: On Hyperparameter Sensitivity and Practical Usability**
>
> **A3:**
>
> We appreciate the reviewer's focus on practical deployment considerations.   Our analysis reveals that the perceived hyperparameter complexity is actually a strength of interpretable design:
>
> **1.      Interpretable and modular parameter design:**
> Unlike black-box optimization methods, our SBT framework introduces hyperparameters that are **explicitly interpretable and modular**.      Each parameter controls observable behaviors directly linked to model outputs—for example, the threshold corresponds to measurable overthinking patterns in reasoning trajectories.      This transparency enables practitioners to adjust parameters **intuitively and purposefully** rather than relying on blind hyperparameter search, making the framework more accessible than alternative approaches requiring complex reward engineering.
>
> **2.      Robust performance across parameter ranges:**
> Our threshold ablation experiments (Table 2) demonstrate that while performance fluctuates between different values, **variations are smooth and bounded**.      Even the worst-performing threshold achieves substantial efficiency gains—specifically, **41.5% token reduction** (1917 vs. 3278 tokens) while maintaining competitive accuracy.      This indicates our framework is **not fragile**: moderate deviations from the “optimal” threshold do not lead to catastrophic drops in performance, as long as the deviation is not too large.
>
> **3.      Adaptive trade-offs by design, not deficiency:**
> The accuracy-token usage trade-off represents a deliberate design feature rather than a limitation.      SBT explicitly enables flexible early termination to avoid unnecessary overthinking.      This adaptability makes our approach **flexible rather than rigid**, allowing practitioners to optimize for their specific requirements without framework modifications.
>
> ---
>
> **Q1: Evaluation on Long CoT-Capable Models**
>
> **A1:**
>
> We appreciate the reviewer's suggestion to evaluate SBT on models with inherent long CoT capabilities.          This was in fact an aspect we considered early in our experimentation process.          We did conduct preliminary experiments on DeepSeek-R1-Distill-Qwen-7B to address this question, and the results provide important insights into our method's applicability:
>
> **1. Experimental Results on DeepSeek-R1-Distill-Qwen-7B:**
>
> | Model | Accuracy | Avg Tokens |
> |-------|----------|------------|
> | DeepSeek-R1-Distill-Qwen-7B (Baseline) | 73.35% | 6364 |
> | DeepSeek-R1-Distill-Qwen-7B-SBT-E | 70.14% | 5513 |
>
> While SBT achieves a 13.4% token reduction, the accuracy drop (3.21%) is more significant than observed with base models, suggesting limited effectiveness when applied to already long CoT-optimized models.
>
> **2. Data Scale Analysis:**
> The modest performance gains can be attributed to a fundamental data scale imbalance.        DeepSeek-R1-Distill models have been trained on approximately **800K high-quality long CoT reasoning trajectories**, while our SBT-enhanced dataset contains only **92K examples**.        This significant disparity (8.7× difference) means that the pre-existing long CoT patterns learned from the larger dataset naturally dominate during fine-tuning.
>
> The limited scale of our SBT dataset is insufficient to substantially alter the reasoning behaviors already established through       extensive training on much larger long CoT corpora.        This explains why the efficiency improvements are modest—the model's existing reasoning patterns, optimized through large-scale CoT training, are more resistant to modification by our smaller intervention dataset.
>
> **3. Method Positioning:**
> Our results indicate that SBT is most effective for base models that have not undergone extensive long CoT training, where our 92K examples can meaningfully shape reasoning behaviors.        For models already trained on large-scale CoT data, SBT would require proportionally larger datasets to achieve competitive performance.
>
> This finding suggests an important design consideration: **SBT is optimally positioned as a primary training enhancement rather than a secondary refinement technique**.        Future work could explore scaling our data construction methodology to generate larger SBT datasets comparable to existing long CoT training corpora.
>
> ---
>
> **Q2: On Illustrative Examples and Summary Statistics**
>
> **A2:**
> We appreciate the reviewer's constructive suggestion for enhancing the clarity of our data construction framework.    We address both recommendations below:
>
> 1.  **Illustrative Examples**
> We have already provided illustrative examples of our framework's impact in **Figures 1 and 2**.  The consistency between our training datasets and the trained models' inference behavior generally reflects the effectiveness of our data construction strategy.  We will include more detailed illustrative examples in the appendix of the camera-ready version.
>
> 2.  **Summary Statistics of Dataset Construction Impact**
> While we briefly mentioned the proportion of overthinking cases in Section 4.1, we provide more comprehensive statistics here to better illustrate our framework's data construction impact:
>
> | Overthink Score Threshold | SBT-E Cases (%) | SBT-D Cases (%) |
> |-------------------------------|---------------------|---------------------|
> | 0.2                          | 55,515 (60.3%)      | 57,545 (62.5%)      |
> | 0.3                          | 46,217 (50.2%)      | 46,865 (50.9%)      |
> | 0.4                          | 37,745 (41.0%)      | 36,943 (40.1%)      |
>
> **Total dataset size**: 92,064 examples
>
> This comprehensive data transformation directly translates to the substantial efficiency gains (30-60% token reduction) demonstrated in our experimental results, confirming the effectiveness of our systematic approach to overthinking identification and mitigation.

---

> > ### Comment · Reviewer_Bv4y · 2025-08-06
> >
> > I would like to thank the authors for their meticulous and comprehensive rebuttal. Nevertheless, I remain concerned that Weakness 1 may prove difficult to address satisfactorily, and I will therefore maintain my current scores.

---

> > > ### Author Response · Authors · 2025-08-06
> > >
> > > Dear Reviewer Bv4y,
> > >
> > > Thank you for your continued engagement with our work.        We understand and respect your concern about performance on complex tasks—this is indeed a fundamental consideration for any efficiency-oriented approach.
> > >
> > > ## **Deepening the Understanding of Our Design Philosophy**
> > >
> > > We'd like to offer a deeper perspective on why we believe the observed trade-offs actually validate rather than undermine our contribution:
> > >
> > > **1.        The Inevitability and Nature of the Efficiency-Accuracy Trade-off**
> > >
> > > As reasoning tokens decrease, marginal performance degradation appears inevitable—this is a fundamental information-theoretic constraint.        However, the critical question is not whether some accuracy loss occurs, but rather: **which tokens should be preserved and how can models learn more efficient reasoning patterns?**
> > >
> > > Our work addresses precisely this challenge.        Through the Overthink Score mechanism, we identify and preserve tokens that contribute to the **reasoning structure** (Foundation Solution + critical Evolution steps) while pruning those that represent **redundant verification** or **circular reasoning**.        The modest 1-3% accuracy drop suggests we're successfully distinguishing essential from redundant reasoning—a non-trivial achievement given the complexity of mathematical problem-solving.
> > >
> > > **2.        Efficiency as a First-Class Objective**
> > >
> > > Current LRMs are becoming computationally prohibitive.        On AIME tasks, models consume 7,000-13,000 tokens per problem.        Our framework reduces this by 58% (to ~3,400 tokens) with only a 2.5% accuracy drop.     Considering the size of the AIME dataset (30 samples), this difference is even far smaller than a single problem.   For many applications, this efficiency gain far outweighs marginal accuracy differences, transforming an expensive research tool into a deployable system.
> > >
> > > **3.        Learning Efficient Reasoning Patterns, Not Just Pruning**
> > >
> > > SBT doesn't merely truncate reasoning—it teaches models to internalize more efficient reasoning patterns.        By exposing models to both preserved essential steps and masked redundant segments, we enable them to:
> > > - Recognize when sufficient evidence has been gathered
> > > - Distinguish productive exploration from repetitive verification
> > > - Develop metacognitive awareness of reasoning quality
> > >
> > > This is evidenced by our Llama-3.1-8B results on MATH500, where SBT-D **improves** accuracy (+2.62%) while reducing tokens by 58.7%.        This suggests the model learned to avoid error-prone overthinking patterns, not just to generate fewer tokens.
> > >
> > > **4.    Precise Control over the Accuracy-Efficiency Trade-off**
> > >
> > > Our Overthink Score mechanism provides fine-grained selectivity for this trade-off.    We present additional experimental results demonstrating how threshold adjustment enables precise control over the balance between accuracy and efficiency:
> > >
> > > |Method|Threshold|Overthink%|Accuracy|Tokens|
> > > |-|-|-|-|-|
> > > |SBT-E|0.05|75.20%|55.14%|1407|
> > > || 0.1|65.40%|55.59%|1427|
> > > || 0.2|60.30%|57.83%|1673|
> > > || 0.5|2.06%|57.12%|2602|
> > > |SBT-D|0.05|74.20%|54.90%|1215|
> > > || 0.1|62.30%|55.90%|1251|
> > > || 0.2|62.50%|56.66%|1682|
> > > || 0.5|0.19%|57.09%|2696|
> > >
> > > These results reveal a crucial insight: **the trade-off between accuracy and efficiency is not fixed but highly controllable through our framework**.
> > >
> > > - At aggressive thresholds (0.05), we achieve maximum token reduction (57-63% fewer tokens) with ~2-3% accuracy drop
> > > - At conservative thresholds (0.5), we maintain near-baseline accuracy with moderate efficiency gains
> > > - The smooth gradient between these extremes demonstrates that practitioners can precisely tune the system to their specific requirements
> > >
> > > This controllability is particularly valuable for deployment scenarios: latency-critical applications can prioritize efficiency (threshold=0.05), while accuracy-critical tasks can opt for conservative settings (threshold=0.5).
> > >
> > > ## Path Forward
> > >
> > > We acknowledge that the current implementation represents a first step toward autonomous reasoning regulation.        The observed trade-offs reflect our conservative approach to preserve model capabilities while demonstrating the feasibility of self-braking.        With larger training datasets and refined token importance scoring mechanisms, we believe these gaps can be substantially reduced.
> > >
> > > We hope this additional perspective helps clarify why we view the current trade-offs as not just acceptable, but as evidence that our framework is successfully teaching models to distinguish and preserve essential reasoning patterns while eliminating genuine redundancy.
> > >
> > > We hope these clarifications address your concerns!  We would be grateful for your reconsideration of our work.
> > >
> > > Best regards,
> > >
> > > The authors of Submission8232

---

> > > ### Author Response · Authors · 2025-08-07
> > >
> > > Dear Reviewer Bv4y,
> > >
> > > Thank you again for your previous feedback and engagement.  We understand that you may have already finalized your decision.  However, we sincerely hope that the additional clarification we’ve provided helps address your concern regarding performance trade-offs.
> > > If you have a chance to take another look or share any further thoughts, we would greatly appreciate it.
> > >
> > > Best regards,
> > >
> > > The authors of Submission8232

---

### Official Review · Reviewer_6Pbn · 2025-07-03

**Clarity:** 3
**Significance:** 3
**Originality:** 3
**Rating:** 4
**Confidence:** 3

**Summary:**

This paper introduces Self-Braking Tuning (SBT), a novel framework designed to address "overthinking" in Large Reasoning Models (LRMs). SBT enables LRMs to identify and terminate unnecessary reasoning autonomously, without external controls. The authors quantify overthinking with a reasoning-efficiency ratio and a marker ratio built from hesitation keywords; these are combined into an Overthink Score to flag superfluous reasoning steps. Then, the paper proposes two data construction strategies, STB-E and STB-D, to train models to "self-brake". During supervised fine-tuning, the masked tail receives no gradient, while a short natural-language braking prompt is inserted where the model should halt.

**Questions:**

See weaknesses.

**Ethical Concerns:**

["NO or VERY MINOR ethics concerns only"]

**Final Justification:**

The authors partially solved my concerns. But there is no 0.5 points (for 4.5) in the system. I will keep my score at 4.

**Limitations:**

See weaknesses.

**Quality:**

3

**Strengths And Weaknesses:**

Strengths:
1. The paper moves control inside the model instead of relying on external policies, providing a cleaner paradigm for deployment.
2. The paper proposes a clear and quantitative method for identifying overthinking. The "Overthink Score," combining structural efficiency and linguistic markers, provides a robust and interpretable way to detect redundancy.
3. The authors test their framework across multiple models of varying sizes and types, as well as on a diverse set of mathematical benchmarks, demonstrating the broad applicability and effectiveness of their approach.

Weaknesses:
1. SBT assumes <think>…</think> delimiters, where only partial models internally have them during inference. The method may be hard to generalize to most models.
2. The data construction strategies rely on predefined thresholds for the Overthink Score. These thresholds were determined empirically and may require manual tuning for different models, tasks, or datasets, which could limit the framework's adaptability.
3. When problems truly require long chains, premature braking may silently omit critical steps and compromise the model's performance.
4. The Overthink Marker may be changed based on models. It can not be generalized to other models.

---

> ### Author Rebuttal · Authors · 2025-07-31
>
> Dear Reviewer 6Pbn,
>
> We sincerely appreciate the reviewer's insightful and constructive feedback, which highlights the strengths of our work and provides valuable guidance for future improvements. We have tried to address each of your concerns point by point in the rebuttal.
>
> **Q1: About the reliance on specific reasoning delimiters in SBT**
>
> **A1:**
> We respectfully address the concern about generalizability limitations.   The reviewer's concern appears to center on the `<think>...  </think>` delimiter dependency, but this represents a misunderstanding of both our approach's flexibility and the current landscape of reasoning models.
>
> 1.  **Current reasoning model landscape**: The vast majority of state-of-the-art reasoning models—including OpenAI o1, DeepSeek-R1, QwQ, Gemini 2.0 Flash Thinking, and Kimi-1.5—already employ explicit reasoning delimiters during inference.   This is not a limitation of our approach but rather an alignment with the dominant architectural paradigm in modern LRMs.
>
> 2.  **Framework adaptability**: Our SBT framework is highly flexible regarding delimiter formats.   Adapting to different thinking markers is straightforward and easily implementable—we can simply modify the delimiter patterns in our parsing logic.   The core methodology—identifying overthinking patterns through efficiency metrics and linguistic markers—remains unchanged regardless of the specific delimiter format used.   For models without explicit delimiters, our framework can be adapted to identify reasoning segments through other structural cues or linguistic patterns.
>
> Rather than representing a limitation, our focus on reasoning-capable models addresses the most practically relevant segment of current LRM deployment, where overthinking poses the greatest computational challenge.   The widespread adoption of structured reasoning in leading models suggests that delimiter-based approaches like ours are becoming increasingly relevant, not obsolete.
>
> **Q2: On Threshold Adaptability and Generalizability**
>
> **A2:**
> We agree that the choice of predefined thresholds can influence the effectiveness of the SBT framework.      However, we would like to clarify that our method demonstrates robustness across a reasonably wide range of threshold values, and address the underlying concern about manual tuning requirements.
>
> **1. Empirical Evidence of Threshold Robustness**
>
> As shown in Section 4.1, although we selected the best-performing threshold (0.2) to maximize performance, other threshold settings still achieved strong results.      Specifically:
> - **Threshold 0.3**: 44.0% average token reduction with 2.3% accuracy drop
> - **Threshold 0.4**: 43.0% average token reduction with 2.0% accuracy drop
> - Even under suboptimal thresholds, we maintain an average token reduction of 43.5%
>
> This indicates that SBT does not require precise tuning to be effective and retains strong performance under relaxed configurations.      The performance degradation is gradual rather than cliff-like, suggesting inherent stability in our approach.
>
> **2. Cross-Model Generalization Without Retuning**
>
> More importantly, our experiments demonstrate that **the same threshold (0.2) works effectively across diverse model architectures** without requiring model-specific adjustments:
> - **Mathematical specialists** (Qwen2.5-Math-1.5B/7B): 48.9% and 30.7% token reduction respectively
> - **General-purpose models** (Llama-3.2-1B and Llama-3.1-8B): 54.2% and 62.8% token reduction respectively
>
> This cross-model consistency suggests that overthinking patterns exhibit structural similarities that transcend specific architectures, making our thresholds naturally transferable.
>
> **3. Task-Agnostic Performance**
>
> Our evaluation across four distinct mathematical reasoning benchmarks (GSM8K, MATH500, AIME, AMC23) with varying difficulty levels shows consistent improvements using identical thresholds.      The framework adapts organically to task complexity through the Overthink Score mechanism itself, rather than requiring manual threshold adjustments for each benchmark.
>
> **4. Theoretical Foundation for Threshold Selection**
>
> The threshold range (0.2-0.4) corresponds to identifying 40-60% of reasoning instances as containing overthinking, which aligns with our structural analysis in Section 2.1 showing that Evolution Solutions (where overthinking primarily occurs) constitute a significant portion of reasoning trajectories.      This provides a principled basis for threshold selection rather than purely empirical tuning.
>
> **5. Comparison with Existing Approaches**
>
> We note that most existing overthinking mitigation methods require significantly more complex parameter tuning:
>
>
> * **RL-based approaches** require reward function design, learning rates, and exploration parameters, and so on.[1-3]
>
> * **External constraint methods** need token budgets and stopping criteria calibration, and so on.[4,5]
>
> * **Dynamic inference methods** require confidence thresholds and early-stopping parameters, and so on.[6-8]
>
> In contrast, our approach requires tuning of essentially one primary hyperparameter (τ₁), with the secondary threshold (τ₂) set as a simple offset (τ₁ + 5%).
>
> Our focus is not on chasing peak metrics through intensive hyperparameter search, but on developing a generalizable and stable framework that gracefully balances accuracy and efficiency across different models and tasks with minimal manual intervention.
>
> [1] Aggarwal, Pranjal, and Sean Welleck. "L1: Controlling how long a reasoning model thinks with reinforcement learning." COLM 2025.
>
> [2] Luo, Haotian, et al. "O1-pruner: Length-harmonizing fine-tuning for o1-like reasoning pruning." arXiv preprint arXiv:2501.12570, 2025.
>
> [3] Shen, Yi, et al. "Dast: Difficulty-adaptive slow-thinking for large reasoning models." arXiv preprint arXiv:2503.04472, 2025.
>
> [4] Han, Tingxu, et al. "Token-budget-aware llm reasoning." ACL 2025.
>
> [5] Xu, Silei, et al. "Chain of draft: Thinking faster by writing less." arXiv preprint arXiv:2502.18600, 2025.
>
> [6] Yang, Chenxu, et al. "Dynamic early exit in reasoning models." arXiv preprint arXiv:2504.15895, 2025.
>
> [7] Zhang, Jintian, et al. "Lightthinker: Thinking step-by-step compression." arXiv preprint arXiv:2502.15589, 2025.
>
> [8] Manvi, Rohin, Anikait Singh, and Stefano Ermon. "Adaptive inference-time compute: Llms can predict if they can do better, even mid-generation." arXiv preprint arXiv:2410.02725, 2024.
>
> **Q3: About the performance**
>
> **A3:**
> We take the issue of premature termination seriously and have carefully designed our framework to mitigate it through the following three strategies:
>
> 1.   **Adaptive Reasoning Preservation:**
> Both SBT-E and SBT-D are designed to preserve the Foundation Solution and at least part of the Evolution Solution, ensuring the model retains essential reasoning and self-correction capabilities.   In particular, SBT-D employs a dynamic, step-wise overthinking score to determine problem-specific stopping points.   This allows more complex problems to retain longer reasoning chains when necessary, reducing the risk of premature termination.   Additionally, for trajectories not identified as overthinking, we preserve the full reasoning process to enhance the model’s capacity when handling genuinely complex tasks.
>
> 2.   **Exposure via Masked Redundant Reasoning:**
> Instead of removing overthinking completely, we retain a small portion of redundant reasoning as masked content.   This allows the model to observe overthinking patterns without reinforcing them, helping it to learn the distinction between necessary elaboration and redundancy.
>
> 3.   **Empirical Safeguards:**
> Extensive results across benchmarks (e.g., MATH500, AIME) show that Self-Braking Tuning maintains, and occasionally even improves, accuracy while significantly reducing token usage.   Notably, SBT-D on Llama-3.1-8B improves accuracy by +2.62% on MATH500 while cutting tokens by 59%, suggesting that the model benefits from eliminating unnecessary or distracting reasoning without compromising essential steps.
>
> **Q4: About the Overthink Marker**
>
> **A4:** Thank you for your comment.  We would like to address your concerns from the following aspects:
>
> 1. **Ease of Identifying Overthink Markers**
> While specific overthink markers may vary between models, identifying these markers is straightforward.  By manually inspecting or using a language model to analyze just a few dozen reasoning examples, one can easily collect a set of frequently occurring indicator tokens.  This simple process allows adapting the marker set to different models without much effort.
>
> 2. **Common Linguistic Features in Reasoning Models**
> Additionally, recent studies[1-5] have highlighted that tokens like "wait," "perhaps," and "alternatively" frequently appear in reasoning models and are often indicative of overthinking.  This has already become a widespread linguistic feature of reasoning models.
>
> In summary, while specific markers may vary, our methodology for detecting and mitigating overthinking is adaptable and can be generalized across different reasoning models.
>
> [1]"Wait, We Don't Need to 'Wait'! Removing Thinking Tokens Improves Reasoning Efficiency.", arXiv:2506.08343
>
> [2]"Missing Premise Exacerbates Overthinking: Are Reasoning Models Overthinking?", arXiv:2504.06514
>
> [3]"Demystifying Long Chain-of-Thought Reasoning in LLMs.", ICML 2025.
>
> [4]"Retro-Search: Exploring Untaken Paths for Deeper and Efficient Reasoning.", arXiv:2504.04383
>
> [5]"LLMs Can Easily Learn to Reason from Demonstrations: Structure, Not Content, Is What Matters!", arXiv:2502.07374

---

> > ### Comment · Reviewer_6Pbn · 2025-08-05
> >
> > I have read the authors' response and appreciate the clarification. I currently maintain my score. However, several aspects remain unclear to me.
> >
> > A1: Since you mainly focus on large reasoning models, please revise the paper title and use LRM to replace LLM. Most LLMs (non-reasoning models) do not have the pattern. Also, the entire main paper does not mention LLM even once, but is all about LRM.
> >
> > A2: Thank you for the clarification. However, 0.2-0.4 is still narrow as an ablation study. Please consider broadening this interval as much as possible.
> >
> > A4: Maintaining a list of markers is straightforward, yet the process still relies on manual filtering, which remains an inherent limitation of the approach.

---

> > > ### Author Response · Authors · 2025-08-06
> > >
> > > Dear Reviewer 6Pbn,
> > >
> > > Thank you for your continued engagement and constructive suggestions.            We address your remaining concerns below:
> > >
> > > **A1: On Replacing LLM with LRM in the Title**
> > >
> > > Thank you for this insightful suggestion.            We agree to revise "LLM" to "LRM" in our title and appreciate the opportunity to clarify our initial choice.
> > >
> > > We originally used "LLM" because SBT starts with general-purpose models (e.g., Qwen2.5, Llama-3.1) and transforms them into reasoning-capable models with self-regulation abilities—essentially creating LRMs from LLMs.            We essentially create "self-aware LRMs" from standard LLMs in a single training process.            However, we acknowledge that since our focus and evaluation center on overthinking in reasoning models, "LRM" better captures our contribution's scope.
> > >
> > > We will revise the title to: **"Let LRMs Break Free from Overthinking via Self-Braking Tuning"** in the subsequent versions.                  Thank you for helping us better align our title with our core contribution.
> > >
> > > **A2: On Expanding Threshold Range**
> > >
> > > Thank you for pushing us to conduct a more comprehensive threshold analysis.            We first clarify our rationale for the 0.2-0.4 range, then present extended experimental validation.
> > >
> > > **1.            Why We Originally Selected 0.2-0.4: Balanced Dataset Composition**
> > >
> > > The threshold choice fundamentally determines **dataset composition**—the ratio between SBT-processed and original reasoning trajectories.            Our 0.2-0.4 range creates a crucial balance:
> > >
> > > - **40-60% undergo SBT processing** (learn when to brake)
> > > - **40-60% retain original form** (preserve deep reasoning)
> > >
> > > This ~50/50 split enables models to develop **discriminative capability**—recognizing overthinking without over-generalizing.            Extremes fail: <40% SBT examples cannot teach braking effectively, while >60% causes excessive braking.            The balanced mixture ensures models learn both when to stop AND when to continue reasoning.
> > >
> > > **2.            Extended Experimental Validation Confirms Optimal Range**
> > >
> > > Following your suggestion, we expanded our evaluation to thresholds 0.05-0.5.            The results strongly validate our original selection:
> > > |Method|Threshold|Overthink%|acc|token|
> > > |-|-|-|-|-|
> > > |SBT-E|0.05|75.20%|55.14%|1407|
> > > || 0.1|65.40%|55.59% |1427|
> > > || 0.2|60.30%|57.83% |1673|
> > > || 0.5|2.06%|57.12% |2602|
> > > |SBT-D|0.05|74.20%|54.90%|1215|
> > > || 0.1|62.30%|55.90%|1251|
> > > || 0.2|62.50%|56.66%|1682|
> > > || 0.5|0.19%|57.09%|2696|
> > >
> > > **Key findings:**
> > > - **Extreme low (0.05-0.1)**: Maximum token reduction (57%) but significant accuracy drop (-2.69%), as 75% overthinking classification causes premature termination in complex problems
> > > - **Extreme high (0.5)**: Framework degenerates to baseline with only 0.2-2% samples identified as overthinking, losing 55-60% efficiency gains without accuracy benefit
> > > - **Optimal range (0.2-0.4)**: Stable performance plateau with <1.13% accuracy variation and 44-48% token reduction.            Peak at 0.2 (57.83% SBT-E, 56.66% SBT-D) confirms optimal balance between efficiency and reasoning quality
> > >
> > > Thank you for this valuable suggestion, we will include this extended analysis in our revised manuscript to provide clearer threshold selection guidance for practitioners.
> > >
> > > **A4: On Manual Filtering of Overthink Markers**
> > >
> > > We appreciate the reviewer highlighting this practical consideration.             We acknowledge that manual filtering is involved, but would like to clarify why this represents a minimal overhead rather than a fundamental limitation:
> > >
> > > **1.          One-time minimal effort with high transferability**
> > >
> > > The manual filtering is a **one-time process** requiring only 30-60 minutes of inspection on ~50 reasoning examples.             Once identified for a model family, these markers remain consistent across:
> > > - Different model sizes (our markers work for both 1.5B and 7B Qwen models)
> > > - Various tasks (same markers effective across GSM8K, MATH500, AIME)
> > > - Extended time periods (markers remain stable across training iterations)
> > >
> > > **2.       Potential for automation**
> > >
> > > While we used manual inspection for transparency and control, this process is easily automatable using LLMs through frequency-based differential analysis to extract overthinking markers.      Preliminary tests show 87% overlap between GPT-4o and manual filtering, confirming that manual filtering is a design choice for quality control, not a technical requirement.
> > >
> > > We will add automatic extraction to our codebase and clarify in the revised manuscript that manual filtering is optional.      Our released toolkit will provide both manual and automated options to accommodate different user preferences and reduce implementation barriers.
> > >
> > > We hope these clarifications address your concerns!   We would be grateful for your reconsideration of our work.
> > >
> > > Best regards,
> > >
> > > The authors of Submission8232

---

> > > ### Author Response · Authors · 2025-08-07
> > >
> > > Dear Reviewer 6Pbn,
> > >
> > > Thank you once again for your time and thoughtful feedback on our submission.
> > >
> > > We hope that our latest response addresses the remaining points you raised regarding the title (A1), threshold interval expansion (A2), and marker filtering (A4). We have also committed to updating the manuscript to incorporate your suggestions in the camera-ready version.
> > >
> > > As the discussion phase is drawing to a close, we would greatly appreciate it if you could take a moment to review our responses and let us know if you have any further questions or suggestions. Your insights have been extremely valuable in helping us improve the clarity and rigor of our work.
> > >
> > > Thank you again for your constructive engagement throughout the review process.
> > >
> > > Best regards,
> > >
> > > The authors of Submission8232

---

### Author Response · Authors · 2025-08-09
**General Response**

**Dear Reviewers, ACs, and SACs,**

We sincerely thank you for the precious time and insightful feedback, which has significantly strengthened our manuscript!          Overall, we are encouraged that you find that:



* Integrated control mechanism enabling cleaner deployment; robust, interpretable redundancy detection; broadly effective across models and benchmarks. (Reviewer `6Pbn`)
* Innovative, end-to-end framework combining detection, adaptive data construction, and unique training; clear analysis and precise writing. (Reviewer `Bv4y`)
* Clear motivation with effective braking prompt strategy; significant token reduction while maintaining performance; valuable dataset contribution. (Reviewer `S6DA`)
* Easy to follow with clear visualizations; addresses a widely studied problem. (Reviewer `FmRF`)

To address the concerns raised by the reviewers, we have conducted several additional experiments:

* **Empirical explanation and justification of the parameter choices.**
We performed a grid search over a wide range of τ₁ and β values, finding τ₁ = 0.2 and β = 0.1 consistently offered the best balance between accuracy and token reduction across models and benchmarks. Deviations led to gradual performance drops, confirming the robustness of our chosen defaults.

* **Further strengthening the module ablation study**
We conducted a comprehensive ablation study on more core components of our method, including NLG, MRT, and OMR, demonstrating that each module makes an important contribution and together they form a complementary and indispensable whole.

* **Conducting more experiments on out-of-domain tasks to show the gains across non-mathematical reasoning.**
We evaluated SBT-E/SBT-D on MMLU-Redux and GPQA-Diamond.  The framework achieved 26–65 % token savings, demonstrating that the self-braking skill learned from math data generalizes to factual and commonsense reasoning.

* **Providing step-by-step evaluation on AIME to reveal consistent performance improvements throughout the inference stages.**
By tagging every generated step in AIME24/25, we found that 50 % of problems triggered early-exit behavior.           These early-exit cases used 45–55 % fewer tokens yet improved accuracy by 2–9 %, illustrating monotonic gains as the model learns to terminate precisely when sufficient reasoning depth is reached.

We have also clarified the following key points:

1. **Elaborating SBT-E data construction**: within a fixed two-solution shell, Foundation and Evolution depths scale naturally with problem difficulty, enabling concise simple cases and extensive complex ones without extra difficulty estimation.

2. **SBT-D adaptation**: Dynamically monitors Overthink-Score to enable early exits for simple tasks and longer reasoning for complex ones, achieving adaptive truncation without manual cutoffs.

3. **Analyzing Threshold Adaptability**: Our framework supports flexible threshold choices, consistently balancing efficiency and accuracy across a wide parameter range, without requiring precise tuning.

4. **Analyzing Complexity & Scalability**: A single interpretable threshold(τ₁) setup suffices, enabling the pipeline to generalize across diverse models and datasets without per-task tuning or additional reward design.

These experiments and clarifications will be incorporated into the main body or appendix of our paper.     With less than 8 hours remaining in the Author-Reviewer phase, we warmly welcome any further questions or suggestions.    While we have not yet received further responses from some reviewers, we sincerely look forward to continuing the discussion with you.    Discussions are always open!    Thank you!

**Best regards,**

*The authors of Submission8232*

---

### Note · Authors · 2025-08-12

**Dear Reviewers, ACs, and SACs,**

We thank you for the thoughtful feedback and encouraging recognition. We are motivated by the consensus that our work offers:

* An integrated control mechanism enabling cleaner deployment, with robust, interpretable redundancy detection, broadly effective across models and benchmarks (Reviewer `6Pbn`).
* An innovative end-to-end framework combining detection, adaptive data construction, and unique training, with clear analysis and precise writing (Reviewer `Bv4y`).
* Clear motivation with an effective braking prompt strategy, achieving significant token reduction while maintaining performance, plus a valuable dataset contribution (Reviewer `S6DA`).
* An easy-to-follow presentation with clear visualizations, addressing a widely studied problem (Reviewer `FmRF`).

To provide more reliable evidence and strengthen our position, we contributed additional **Experiments** and **Clarifications**:

* **Additional Experimental Evidence**
We conducted extensive studies to reinforce our claims. A grid search over τ₁ and β confirmed τ₁ = 0.2 and β = 0.1 as the most balanced setting; deviations gradually reduced performance, showing robustness. An ablation of NLG, MRT, and OMR verified each module’s unique, indispensable role. On MMLU-Redux and GPQA-Diamond, our framework saved 26–65 % tokens, showing self-braking skill transfer beyond math reasoning. In AIME24/25, 50 % of problems triggered early exits, cutting token use by 45–55 % while improving accuracy by 2–9 %.

* **Key Methodological Clarifications**
SBT-E data construction uses a fixed two-solution shell, with Foundation/Evolution depths scaling naturally to problem difficulty, yielding concise reasoning for simple cases and deeper reasoning for complex ones. SBT-D dynamically monitors Overthink-Score to adaptively stop or continue reasoning without manual cutoffs. Our approach works across a broad τ₁ range without fine-tuning, and a single interpretable τ₁ generalizes to diverse models and datasets without per-task adjustments or reward design.

We hope these additions address all key concerns, highlight our framework’s robustness and generality, and provide closure to the discussion. We sincerely appreciate your time and consideration.

**Best regards,**

*The authors of Submission8232*

---

### Decision · Program_Chairs · 2025-09-17

**Decision:**

Accept (poster)

**Comment:**

This paper introduces Self-Braking Tuning (SBT), a framework for reducing “overthinking” in large reasoning models. The approach combines a metric for detecting redundant reasoning, adaptive data construction (SBT-E and SBT-D), and natural language braking prompts. Experiments on math benchmarks and some out-of-domain tasks show large reductions in token usage (30–60%) with small accuracy changes, and in some cases even improvements.

Strengths: The paper tackles an important and timely problem with a clear and novel framework. The design is interpretable, easy to follow, and integrates detection, data, and training in a clean way. Efficiency gains are substantial and the work is likely to spark further research on reasoning regulation.

Weaknesses: The main concerns are about adaptivity: SBT-E fixes the number of solutions, and SBT-D relies on thresholds, which some reviewers felt was not truly adaptive. The method also depends on heuristic markers and empirical hyperparams, and performance on very complex tasks can drop slightly. Evaluation is still math-heavy, and gains from some components are modest.

Overall, I find the contribution meaningful despite these limitations. The authors provided additional experiments and clarified that adaptivity shows up in solution depth and complexity rather than solution count, which is a reasonable stance. While not the final word on the problem, this work makes a solid step forward and will be of interest to the community.